# Revisiting Privacy, Utility, and Efficiency Trade-offs when Fine-Tuning Large Language Models

## Abstract

We study the inherent trade-offs in *minimizing privacy risks and maximizing utility, while maintaining high computational efficiency*, when fine-tuning large language models (LLMs). A number of recent works in privacy research have attempted to mitigate privacy risks posed by memorizing fine-tuning data by using differentially private training methods (e.g., DP-SGD), albeit at a significantly higher computational cost (inefficiency). In parallel, several works in systems research have focused on developing (parameter) efficient fine-tuning methods (e.g., LoRA). However, few works, if any, investigated whether such efficient methods, in isolation, enhance or diminish privacy risks.

In this paper, we investigate this gap and arrive at a surprising conclusion: efficient fine-tuning methods like LoRA mitigate privacy-risks similar to private fine-tuning methods like DP-SGD. Our empirical finding contradicts the prevailing wisdom that privacy and efficiency objectives are at odds during fine-tuning. Our finding is established by (a) carefully defining measures of privacy and utility that distinguish between recollecting *sensitive and non-sensitive* tokens in training and test datasets used in fine-tuning and (b) extensive evaluations using multiple open-source language models from Pythia, Gemma, Llama, and Qwen families and different domain-specific datasets.

## 1 Introduction

Large language models (LLMs) have shown proficiency across diverse natural language tasks (Naveed et al., 2023), finding applications in education (Wang et al., 2024), medical chatbots (Thirunavukarasu et al., 2023), and AI assistants (Dong et al., 2023). Their capabilities stem from a two-phase process: (a) pre-training on extensive web data (Kaplan et al., 2020) to develop general language understanding (Brown, 2020), (b) fine-tuning on domain-specific data for specific tasks (Zhang et al., 2023). In both phases, the key challenges involve enhancing *privacy* and *efficiency* while maintaining the models' *utility*. Privacy is related to reducing the risk of LLMs leaking sensitive user information contained in the training data, efficiency is related to reducing the computational cost of training, while utility is related to generative performance over test data.

A long line of recent research in the privacy community has focussed on methods to mitigate privacy risks when training LLMs (Shi et al., 2022; Zhao et al., 2022; Abadi et al., 2016; Yu et al., 2022). A notable example of such methods is differential privacy based stochastic gradient descent (DP-SGD)(Abadi et al., 2016). Simultaneously, a flurry of recent research in the systems community has explored parameter-efficient fine-tuning methods in LLMs (Han et al., 2024). A notable example of this class of methods is low-rank adaptation (LoRA) (Hu et al., 2022). However, no prior works, to the best of our knowledge, have investigated the privacy risks associated with the efficient training methods.

The central question driving our research here is: *do efficient fine-tuning methods enhance or mitigate privacy risks during training?* As DP-SGD incurs significant additional computational overhead (Dupuy et al., 2022), while LoRA significantly reduces the computational costs, the answer to the above question can have significant consequences for achieving good privacy-efficiency-utility tradeoffs when fine-tuning. For instance, if LoRA mitigates privacy risks of training that would

suggest that it can simultaneously achieve both privacy and efficiency objectives, contradicting the conventional wisdom drawn from DP literature that privacy comes at a computational cost. A key (surprising) finding of our work lies in establishing that *LoRA does indeed mitigate privacy risks*.

A conjecture that might explain our finding is rooted in the following high-level observations about DP-SGD and LoRA: methodologically, both DP-SGD and LoRA restrict the impact that training examples can have on model parameters – DP-SGD deliberately through its noisy gradient update, and LoRA through low-rank adaptation. We formalize this intuition in this work.

When attempting to answer the above question, we encountered a more foundational question: *how should one quantify such privacy risks associated with a fine-tuning method, so that it allows for a performance comparison across different methods?* Numerous studies have highlighted privacy risks in LLMs due to their tendency to memorize and regurgitate training data containing sensitive personally identifiable information (PII) such as names, emails, and credentials (Mattern et al., 2023; Fu et al., 2023; Kaneko et al., 2024; Carlini et al., 2021; Mireshghallah et al., 2022b; Panda et al., 2024). A natural way to account for privacy risks from memorization might be to measure loss on recollecting tokens in training data sequences.

However, we find that LLMs exhibit very different losses in recollecting sensitive vs. non-sensitive tokens in training data (see Figure 1). This difference is due to inherent randomness and unpredictability of sensitive data (e.g., phone numbers, SSNs) compared to non-sensitive data (e.g., "The dog chases the _"), which is often more structured and predictable. Consequently, we propose a new privacy measure that explicitly account for this difference: aiming for high loss on sensitive tokens from training data.

**Contributions.** We summarize our main findings:

*1. Quantifying privacy and utility, when training LLMs:* We conceptually argue and empirically demonstrate that LLM's ability to recollect (predict) sensitive and non-sensitive data in training (test) datasets are so starkly different that we need to account for them when quantifying privacy and utility. Privacy is best captured by a model's ability to *recollect sensitive tokens in training data*, while utility is best captured by the model's ability to *predict non-sensitive tokens in test data*.

*2. Comparing privacy-utility-efficiency trade-offs for three different fine-tuning methods:* Our measures allow us to conduct a systematic and extensive empirical study of three different

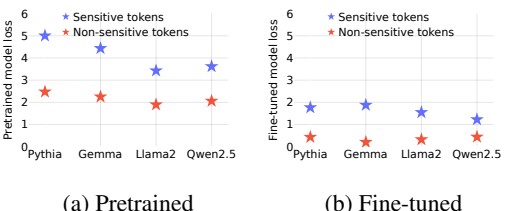

(a) Pretrained       (b) Fine-tuned

Figure 1: Sensitive and non-sensitive tokens have different predictability, measured as the recollection loss by pre-trained models (Figure 1a) and fine-tuned models (Figure 1b). This distinction motivates to quantify privacy using sensitive tokens from the training data.

fine-tuning methods: full fine-tuning (FFT), DP-SGD, and LoRA, using models from four different LLM families, Pythia (Biderman et al., 2023), Gemma (Team et al., 2024), Llama (Touvron et al., 2023) and Qwen (Yang et al., 2024) over two datasets. Our comparative study yields several interesting insights: we find that – FFT results in poor utility-privacy tradeoffs; DP offers reasonable utility-privacy tradeoffs, but is computationally very expensive; LoRA is almost on par with DP in terms of privacy with good privacy-utility tradeoffs, but is more computationally efficient. To reconfirm our results, we evaluate the three methods using existing privacy measures—*privacy loss* (Abadi et al., 2016) and *canary exposure* (Carlini et al., 2019). We also show formally that the effect of fine-tuning a model using LoRA and DP on a datapoint is analogous, leading to privacy benefits.

*3. Feasibility of achieving all the three privacy, utility and efficiency objectives simultaneously:* Our findings about LoRA performance challenge prevailing wisdom that enhancing privacy during training is more computationally expensive. This calls for investigating privacy benefits of existing and new parameter efficient fine-tuning methods.

## 2 BACKGROUND & RELATED WORK

Privacy in large language models (LLMs) is threatened by membership inference attacks (MIAs) and data extraction attacks (DEAs) (Kibriya et al., 2024; Yao et al., 2024; Mireshghallah et al., 2022a; Mattern et al., 2023; Fu et al., 2023; Kaneko et al., 2024; Carlini et al., 2021; Mireshghallah et al., 2022b; Panda et al., 2024), with DEAs posing a more practical risk (Zhang et al., 2024). Differential privacy (DP-SGD) mitigates these risks (Abadi et al., 2016) but is computationally expensive and sometimes vulnerable (Hayes et al., 2023). Parameter-efficient fine-tuning (PEFT) methods (Han et al., 2024) like LoRA (Hu et al., 2022) provide efficiency and utility, though their privacy impact is less explored. We find that LoRA achieves privacy comparable to DP. Unlike prior approaches combining DP-SGD with LoRA (Liu et al., 2023; Yu et al., 2021; Ma et al., 2024), we show all three objectives—privacy, utility, and efficiency—can be achieved with LoRA alone. An extended version of the related work is in Appendix C.

## 3 EXPERIMENTAL SETUP AND METHODOLOGY

In this work, we want to answer the following questions for three different fine-tuning methods – full fine-tuning (FFT), Differential Privacy (DP-SGD), and Low-Rank Adaptation (LoRA):

- *Privacy: How prone is each method to recollecting the sensitive parts of the training data?*
- *Utility : How effective is each method at predicting non-sensitive parts of test data?*
- *Efficiency: What is the computational cost associated with each method?*

We begin by outlining our experimental setup and motivating our definitions of privacy and utility. In Section 4, we study the privacy–utility trade-off in three fine-tuning methods through an exhaustive hyperparameter search. Using the best configurations identified, we then compare the methods via Pareto-optimal trade-offs in Section 5 under both our proposed and existing privacy measures, ensuring a fair evaluation.

We train models from four different families, `Pythia-1B` (Biderman et al., 2023), `Gemma-2B` (Team et al., 2024), `Llama2-7B` (Touvron et al., 2023), and `Qwen2.5-7B` (Yang et al., 2024) for 50 epochs, on two publicly available datasets, `CustomerSim` and `SynBio`. CustomerSim (Shi et al., 2022) is a simulated dialog dataset for conversation with 10k samples, and SynBio (originally called PII) (Holmes et al., 2024) is an LLM generated dataset with student biographies containing PII with $\sim$ 5k samples. We use 80% for training and 20% for evaluation. Table 1 in Appendix D shows excerpts from the datasets. Details on libraries and hyperparameters are provided in Appendix E.

In order to measure privacy and utility, we would need to distinguish between the sensitive and non-sensitive tokens. We define sensitive data as per GDPR Article 4(1) (EU, 2016) which says that it can be any personal data or identifier like name, ID number, or location. Owing to the large scale of data, we leverage two tools for annotating the sensitive information: Presidio (Mendels et al., 2018), which helps in a fast identification of private entities in text, and GPT-4 (Achiam et al., 2023), which is provided with a particular prompt for returning the annotated portions. An example of such a prompt for annotation is provided in Appendix F.

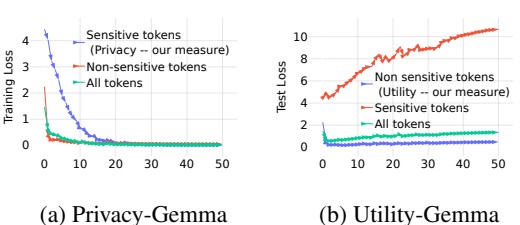

(a) Privacy-Gemma      (b) Utility-Gemma

Figure 2: Our measures offer a precise assessment of privacy and utility by distinguishing between sensitive and non-sensitive tokens.

To confirm the reliability of the annotations at scale, we run a survey among 40 Prolific[1] crowd-workers with a random subset of 100 samples. We provide the details of the survey in Appendix G that depicts that $\sim 75\%$ participants found the GPT-4 annotations to be accurate while Presidio annotations were mostly mixed or under-annotated. *Hence, the rest of the paper has our results using GPT-4 annotations*, and those using Presidio annotations are shown in Appendix I.

---

[1] https://www.prolific.com

Before diving into the tradeoffs from different fine-tuning methods, we want to demonstrate the usefulness of our privacy and utility measures. We quantify **privacy** as the *recollection of sensitive entities in the training data* and **utility** as the *prediction of non-sensitive entities in the test data*.

We present evidence supporting the importance of distinguishing between sensitive and non-sensitive entities in natural language.

**Results.** Figure 2 shows privacy (left) and utility (right) during fine-tuning of the Gemma model on the Customersim dataset (Shi et al., 2022), with training loss on the left column and test loss on the right (results for other models in Appendix H).

In Figure 2a, when no distinction is made between sensitive and non-sensitive tokens, training loss (green curve) starts lower, falsely suggesting strong memorization and higher privacy leakage. Separating losses reveals that *sensitive tokens (blue curve) show much higher loss than non-sensitive ones (red curve)*, especially early in training, since they are less predictable. Similarly, in Figure 2b, test loss without distinction (green curve) appears higher, giving the impression of poor generalization. However, separating losses shows non-sensitive tokens (blue curve) achieve lower loss, confirming better utility. Using these measures, we observe the tradeoffs among different fine-tuning methods.

## 4 PRIVACY-UTILITY-EFFICIENCY INTERPLAY

In this section, we study the impact of privacy, utility, and efficiency on three different fine-tuning methods - full fine-tuning, differential privacy and LoRA. We conduct an extensive hyperparameter search for both differential privacy and LoRA, systematically tuning their respective parameters to explore the privacy–utility trade-off in depth.

**Update rules:** For each fine-tuning method, we describe how updated weights $W_{t+1}$ are computed from the previous weights $W_t$ in each step, where $W_0$ are the weights of the pre-trained base model before fine-tuning. We use $X$ to refer to a batch of $|X|$ data-points and $x_i$ to refer to individual datapoint, $\mathcal{M}_W$ to refer to the model parameterized by weights $W$, and $\mathcal{L}(\mathcal{M}_W(X), X)$ to refer to the autoregressive cross-entropy loss of the model on data $X$. We denote by $\nabla_W \mathcal{L}(...)$ as the gradient of the loss wrt. weights $W$ and $\eta$ is the learning rate.

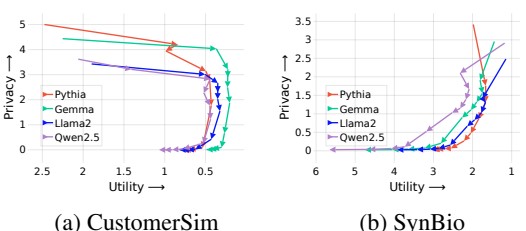

(a) CustomerSim      (b) SynBio

Figure 3: Privacy-utility trade-off shows that privacy increases with higher training loss on sensitive tokens, while utility improves with lower test loss on non-sensitive tokens, enabling desired checkpoint selection to balance both objectives.

**Efficiency:** The efficiency of each method is determined by the amount of computation it requires, and also other factors such as memory requirements. Following (Kaplan et al., 2020), we estimate the amount of training compute ($C$) in floating point operations (FLOPs) for full fine-tuning as $C_{\text{FFT}} = 6DN$, where $D$ is the number of training tokens and $N$, the number of model parameters. For each method, we report its compute requirements relative to the FFT-baseline based on measurements using the PyTorch profiler[2].

### 4.1 FULL FINE-TUNING FOR UTILITY COSTS PRIVACY

**Update rules:** Full fine-tuning (FFT) updates all model parameters at each step:

$$W_{t+1} = W_t - \eta \nabla_{W_t} \mathcal{L}(\mathcal{M}_{W_t}(X), X) \tag{1}$$

**Privacy-Utility trade-off:** Figures 3a and 3b show the *privacy-utility trade-off* for the Customer-Sim and SynBio datasets, respectively. *Privacy increases* with *increasing training loss on sensitive tokens* (up $\Uparrow$ on the y-axis), while *utility increases* with *decreasing test loss on non-sensitive tokens* (right $\Longrightarrow$ on the x-axis). Each curve starts with the baseline performance of the pre-trained model.

---

[2]https://pytorch.org/docs/stable/profiler

For CustomerSim (Figure 3a), as training advances (*denoted by an arrow* → *on the lines*), privacy progressively decreases (*lower on the y-axis*), while utility improves (*rightward on the x-axis*) for approximately the first 5 epochs across all models before stabilizing and eventually declining (*leftward on the x-axis*). However, for SynBio (Figure 3b), the privacy-utility trade-off primarily worsens for Gemma and Llama models. On examining these curves, one can select a desired checkpoint that aligns with specified privacy and utility thresholds.

**Efficiency:** FFT serves as our efficiency baseline. It has moderate compute requirements (discussed above), and relatively high memory requirements, since in addition to the input-dependent activations, we need to keep four numbers per model parameter in GPU memory: the parameter value, its gradient, and two optimizer states (first and second moments for Adam (Kingma, 2014)).

**Takeaway:** FFT offers *poor privacy-utility* trade-offs, since gains in utility in most cases come at the cost of a significant loss in privacy. During FFT, models learn to predict the training distribution better, but also quickly learn to recollect sensitive tokens. FFT is *moderately efficient* and has relatively high memory requirements.

## 4.2 DP-SGD FINE-TUNING FOR PRIVACY AND UTILITY COSTS EFFICIENCY

Differential Privacy (DP-SGD) algorithms (Abadi et al., 2016) aim to safeguard the privacy of individual training data points by limiting their influence on the gradient updates during training. While DP-SGD is designed for MIAs, we show its privacy benefits even in the case of DEAs.

**Update rules:** DP-SGD clips the $l_2$ norm of each data point's gradient at a threshold $T$, followed by adding noise with magnitude $\sigma$ to each clipped gradient. Clipping helps to reduce data sensitivity by ensuring that data points with high gradients have limited impact on the parameters. Adding noise further obscures the contribution, making it difficult to infer specific data points from the model.

$$W_{t+1} = W_t - \eta \, \text{Noise}\left(\frac{1}{B}\sum_i \text{Clip}\left(\nabla_{W_t}\mathcal{L}(\mathcal{M}_{W_t}(x_i), x_i)\right)\right); \text{Clip}(y) = y/\max\left(1, \frac{\|y\|_2}{T}\right)$$

$$\text{Noise}(y) = y + \mathcal{N}(0, \sigma^2 T^2 \mathbb{1}) \tag{2}$$

Between the two hyperparameters, $\sigma$ and $T$, we vary the noise hyperparameter $\sigma$ for the experiments. We found that fixing $\sigma$ to 0.1, and varying $T$ from 0 to 1 was not creating a significant change in privacy. Hence, we resorted to a value of $10^{-2}$ as used in (Shi et al., 2022).

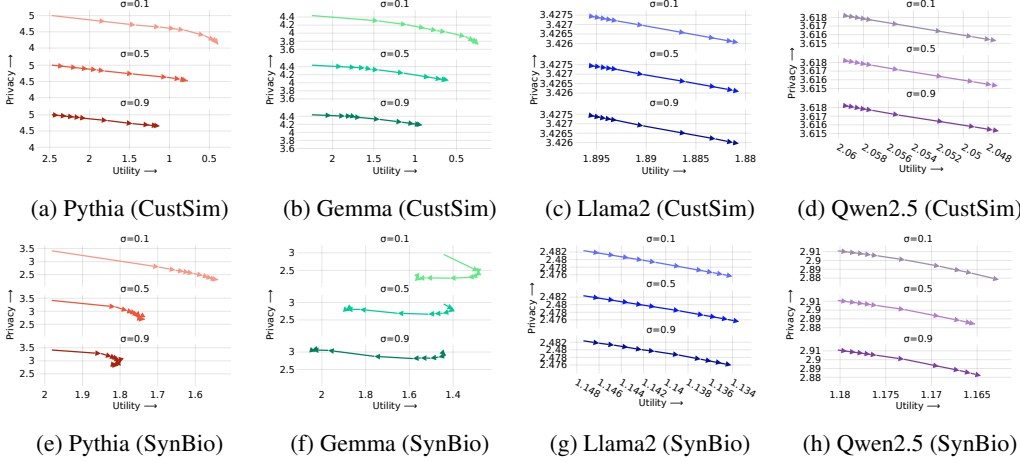

(a) Pythia (CustSim)  (b) Gemma (CustSim)  (c) Llama2 (CustSim)  (d) Qwen2.5 (CustSim)

(e) Pythia (SynBio)  (f) Gemma (SynBio)  (g) Llama2 (SynBio)  (h) Qwen2.5 (SynBio)

Figure 4: Privacy–utility trade-offs for differential privacy (DP-SGD) across models and datasets with varying noise levels ($\sigma$). Larger $\sigma$ increases privacy but reduces utility, with the decline more pronounced in larger models. On SynBio, DP-SGD exhibits a similar pattern. Gemma shows a unique utility drop after two epochs, highlighting the complexity of the privacy-utility trade-off.

**Privacy-Utility trade-off:** Figures 4a - 4d illustrate the privacy-utility trade-off when varying the noise $\sigma$ on the CustomerSim dataset across the models. Similar to previous plots, each curve begins with the performance of the pre-trained model. It is evident that DP-SGD maintains privacy effectively with minimal degradation, but with a trade-off between privacy and utility. For all models, lower noise value of $\sigma = 0.1$ achieves better utility while higher values of $\sigma = 0.5$ , $0.9$ provide better privacy. The total amount of utility achievable with DP-SGD is limited for the larger models like Llama2 and Qwen2.5, possibly due to clipping of larger norms that leads to suboptimal performance (Shen et al., 2021; Tramer & Boneh, 2020). Similar trends can be observed for SynBio in Figures 4e, 4g,4h, but for Figure 4f (with Gemma) utility declines after approximately two epochs.

**Efficiency:** Differential privacy comes with a high computational cost, since the gradients of each datapoint need norm, clipping computations, and noise additions. Empirically, we observe a relative FLOPs requirement of $C_{\text{DP}}/C_{\text{FFT}} = 1.33$. Additionally, the per-sample operations for clipping mean that we need to keep copies of the gradient for each datapoint in memory, which requires substantially more GPU memory decreasing the feasible batch size and the overall training throughput.

**Takeaway:** DP-SGD offers the *the best privacy-utility trade-off* for small-scale models (Pythia,Gemma) , but the privacy gains come at the *cost of efficiency*. DP-SGD finds it difficult to achieve good utility in larger models(Llama2,Qwen2.5) as also observed in (Shen et al., 2021; Tramer & Boneh, 2020).

### 4.3 LoRA FINE-TUNING FOR EFFICIENCY AND UTILITY IMPROVES PRIVACY

LoRA (Hu et al., 2022) is a parameter-efficient fine-tuning method developed to reduce the compute and memory requirements of fine-tuning LLMs.

**Update rules:**

$$W_{t+1} = W_0 + \frac{\alpha}{r}\Delta W_{t+1}; \Delta W_{t+1} = \Delta W_t - \eta\nabla_{\Delta W_t}\mathcal{L}(\mathcal{M}_{W_t}(X), X) \tag{3}$$

LoRA freezes the weights of the pretrained base model $W_0$ and only fine-tunes an adapter matrix $\Delta W_t$. During training, $\Delta W_t$ is stored separately from $W_0$ as two low-rank matrices with rank $r$: $\Delta W_t = B_t A_t$ with $B_t \in \mathbb{R}^{d\times r}, A_t \in \mathbb{R}^{r\times k}, W_0 \in \mathbb{R}^{d\times k}$. $r$ is typically very small, often between $4$ and $32$. $\alpha$ is a scaling factor for the LoRA adapters affecting the behavior of the base weights $W_0$.

**Privacy-Utility trade-off:** We investigate the effects of varying both the rank $r$ and scaling parameter $\alpha$ of LoRA. We use common rank values of $16$ and $32$ with $\alpha \in \{16, 32, 64, 128\}$.While LoRA has been explored for privacy in conjunction with DP-SGD (Yu et al., 2022), *there has been no prior work that examines the privacy benefits of LoRA alone*.

Figures 5a- 5d show the trade-off with rank 16 and varying parameters of $\alpha$ for the CustomerSim dataset across all the models. As seen in Figures 5a and 5b, smaller-scale models (Pythia and Gemma) exhibit a better privacy-utility trade-off when the rank and $\alpha$ values are equal, compared to the other configurations. For the larger models, the privacy declines at later epochs, while utility is mostly retained. We also analyze the trade-off for the SynBio dataset in Figures 5e- 5h for rank 16 which have similar observations. Similar trends are observed for rank 32 shown in Appendix I. However, relative to rank 16, rank 32 exhibits reduced privacy while maintaining similar utility.

**Efficiency:** LoRA has negligible additional FLOPs required for the low-rank matrices $\Delta W$, though they are much smaller than the full base weight $W_0$ ($r \ll min(d, k)$). However, during the backward pass, LoRA requires much less compute. We observe a relative FLOPs requirement of $C_{\text{LoRA}}/C_{\text{FFT}} \approx 0.65$. The original paper (Hu et al., 2022) reports a 25% speedup during training. LoRA has needs of less GPU memory than FFT, since no optimizer states and gradients need to be stored for the base weights $W_0$, which makes it possible to run it with larger batch-sizes and thus an overall increased training throughput.

**Takeaway:** Ours is the first work to investigate the privacy benefits of LoRA, in isolation. The optimal privacy-utility trade-off is mostly achieved with $r = \alpha$. Finally, LoRA is much more *computationally efficient than FFT and especially DP-SGD*. Overall it provides the *best privacy-utility trade-offs* and shows that *privacy can be achieved without additional computational costs*.

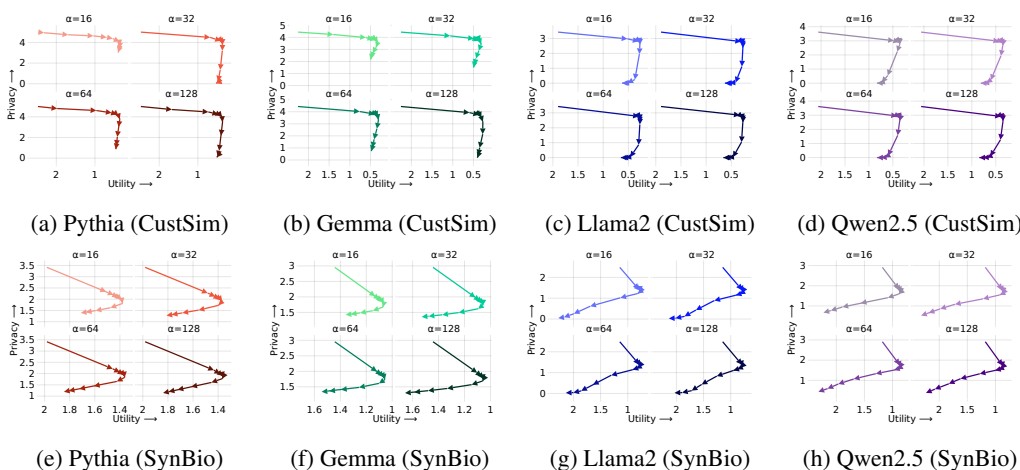

Figure 5: Privacy–utility trade-offs for LoRA fine-tuning with rank 16 and varying scaling factor $\alpha$. Increasing $\alpha$ generally improves or maintains utility but reduces privacy. Smaller models achieve more favorable privacy–utility trade-offs, while larger models retain utility at the cost of reduced privacy during extended training.

## 5 COMPARISON OF FINE-TUNING METHODS

In this section, we compare all fine-tuning methods along three key dimensions—privacy, utility, and efficiency—using the best hyperparameter configurations identified in Section 4. We present Pareto-optimal curves under a fair setting, where each method is evaluated at its best configuration. By examining these curves, one can select the checkpoint most appropriate for the task at hand.

Recall that *utility is measured as test loss on non-sensitive tokens*. Besides measuring privacy as the *training loss on sensitive tokens*, we define two more metrics for privacy : (a) Privacy Loss over sensitive tokens (Abadi et al., 2016) and (b) Canary exposure (Carlini et al., 2019).

**Privacy Loss**: Let $\mathcal{D}$ be the dataset with the sensitive token $d$ under consideration, and $\mathcal{D}'$ be the dataset without it. Considering $\mathcal{M}(\mathcal{D})$ as the fine-tuned model that has seen the datapoint $d$, $\mathcal{M}(\mathcal{D}')$ as the model that has not seen $d$, and leveraging the definition from (Abadi et al., 2016), we can define the privacy loss (PL) as :

$$PL = log\frac{P(\mathcal{M}(\mathcal{D}) = d)}{P(\mathcal{M}(\mathcal{D}') = d)} = log\frac{P_{\mathcal{D}}(d)}{P_{\mathcal{D}'}(d)} \quad (4)$$

A natural approximation of such a model with the data unseen is the pretrained model. Recall that negative log-likelihood for a token $d$ is $NLL_{(D)}(d) = -logP_{\mathcal{D}}(d)$. We may rewrite the privacy loss as the difference between the negative log-likelihood of the pretrained model and the fine-tuned model over the token $d$ :

$$PL = log\frac{P_{\mathcal{D}}(d)}{P_{\mathcal{D}'}(d)} = logP_{\mathcal{D}}(d) - logP_{\mathcal{D}'}(d) = NLL_{\mathcal{D}'}(d) - NLL_{\mathcal{D}}(d) \quad (5)$$

Unlike our measure where we observe the absolute loss on sensitive tokens from the training data, this metric observes the relative change in loss w.r.t a model that has never seen the datapoint.

**Canary exposure**: Originally proposed by (Carlini et al., 2019) to measure unintended memorization, canaries are random sequences (e.g., $s=$"My ID is ○○○○○○") inserted into training data. Exposure is computed by enumerating all possible sequences from the randomness space $\mathcal{R}$ and calculating the negative log-rank. Following the definition of exposure in (Carlini et al., 2019), the exposure of a canary $s[r]$ in a model $\mathcal{M}$ over randomness space $\mathcal{R}$ is defined as:

$$exposure_{\mathcal{M}} = log_2|\mathcal{R}| - log_2rank_{\mathcal{M}}(s[r]) \quad (6)$$

where the rank of a canary is its index in the list of all possibly-instantiated canaries, ordered by the model perplexity of all those sequences. In this setting, we inserted the canary "My ID is 34175" into the training data for 10 times and measured its exposure as a metric for privacy.

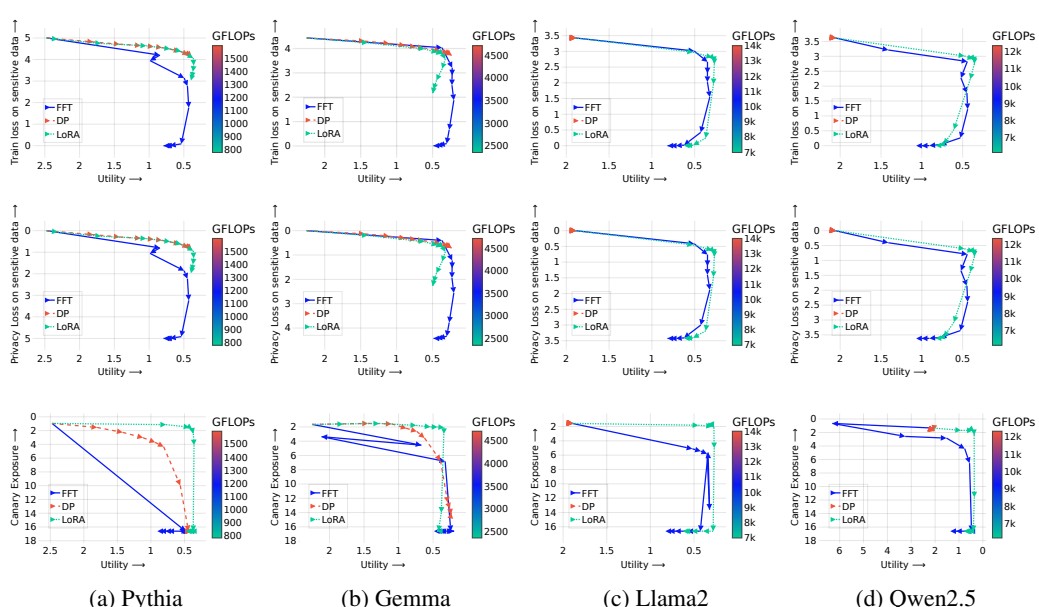

Figure 6: Pareto-optimal tradeoffs (*top-row* : Training loss on sensitive tokens ; *middle row* : Privacy Loss on sensitive tokens ; *bottom-row* : Canary exposure). DP-SGD achieves best privacy-utility tradeoffs on small models with reduced utility on large models, albeit with a significantly high computational cost; FFT offers reasonable efficiency-utility tradeoffs with the worst privacy; and LoRA provides a balanced trade-off among all objectives.

For efficiency, we measure floating point operations (FLOPs) based on the number of operations incurred (e.g., matrix multiplication, addition, etc.) during training.

We obtain the best configuration for DP-SGD with noise ratio of $\sigma = 0.1$, and for LoRA with $r = \alpha = 16, \sigma = 16$. Figure 6 presents the pareto-optimal curves for CustomerSim, comparing the fine-tuning strategies over all the metrics.

**Privacy**: Regarding *privacy*, *FFT* shows *poor privacy* over extended training over all 3 metrics, while *DP* achieves the *highest privacy levels*. *LoRA* provides *similar privacy as DP* throughout most epochs but declines gradually with extended training, especially on the large-scale model.
**Utility:** *FFT* maintains *relatively strong utility* on CustomerSim. *DP-SGD* while *yielding good utility in smaller models* (Pythia,Gemma), *performs poorly in the larger models* (Llama2,Qwen2.5). In contrast, *LoRA consistently preserves higher utility* across the entire training period.
**Efficiency**: The color bar in Figure 6 highlights the FLOPs intensity associated with each fine-tuning strategy. *DP-SGD* requires the *highest number of FLOPs* (in red) due to the need for per-sample gradient computation, where each sample corresponds to a token within each training sequence. *FFT* demands a *moderate number of FLOPs* (in blue), proportional to the total number of parameters. Finally, *LoRA* requires the *fewest FLOPs* (in green), as especially during backpropagation, most operations only operate on the low-rank matrices.

Assessing all the above aspects, this challenges the belief that privacy comes at the cost of efficiency, showing that ***LoRA, an efficient fine-tuning method can provide privacy benefits***. Alongside, LoRA-tuned models retain performance on benchmark datasets similar to pre-trained models throughout training as shown in Figure 28. For a full comparison of methods, refer to Appendix J.

## 6 CONJECTURE: LORA'S PRIVACY BENEFITS RELATIVE TO DP-SGD

In this section, we present a formal conjecture on LoRA's privacy benefits, drawing an analogy to differential privacy (DP-SGD).

**Analogy between LoRA and DP-SGD:** We hypothesize that LoRA's low-rank constraint helps in preserving privacy of the sensitive data. By forcing all the parameter updates into a low-rank subspace, LoRA can reduce the influence a training point can have on the model. Essentially, this is the same quantity that DP-SGD also aims to reduce by clipping and adding noise to the gradients. DP-SGD follows the route of adding randomness, and LoRA follows the route of compressing updates. The effect becomes more on the sensitive data as they are rarely occurring in the training set, which may possibly lead to it being absent in the subspace learnt by LoRA. Empirically, Figure 7 shows that low-rank LoRA model results in better privacy than its higher rank counterparts or full fine-tuning. We formally show the analogy in the theorem below:

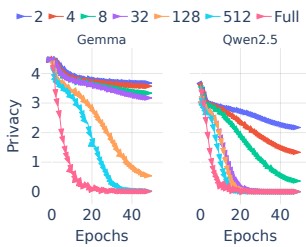

Figure 7: Privacy improves with low-rank matrix.

**Theorem 1.** *Let $D$ and $D'$ be two neighbouring datasets differing by a datapoint $i$. Both LoRA and DP-SGD reduce the influence of $i$ in comparison to a fully trained model, i.e. $\Delta_{DP}^i \leq \Delta_{FFT}^i$ and $\Delta_{LoRA}^i \leq \Delta_{FFT}^i$. This indicates privacy preservation of datapoint $i$ as less the influence, less will be its distinguishability.*

*Proof.* Consider we have $D$ and $D'$ as two neighbouring datasets differing by a datapoint $i$. Let $W(D)$ and $W(D')$ be the respective model parameters where $W(.) \in \mathbb{R}^{d \times k}$. Similarly, let $\Delta W(D)$ and $\Delta W(D')$ be the change in respective aggregate model gradients. Since $D$ and $D'$ differ by a point $i$, we can attribute the difference between $\Delta W(D)$ and $\Delta W(D')$ to the contribution or influence of datapoint $i$ on the aggregate update.

$$\therefore \Delta_{FFT}^i = ||\Delta W(D) - \Delta W(D')||_F = ||g_i||_F$$

For DP-SGD, it is well established that noise (of scale $\sigma$) gets added to the clipped gradients (gradients clipped by a threshold of $T$) that masks the impact of the datapoint (Abadi et al., 2016). We can thereby frame the influence of the datapoint $i$ w.r.t DP-SGD model as:

$$\Delta_{DP}^i = \frac{||g_i||_F}{max(1, \frac{||g_i||_F}{T})} + \mathcal{N}(0, \sigma^2 \mathcal{I}) \approx \frac{||g_i||_F/T}{\sigma} = \frac{||g_i||_F}{T\sigma} \leq \Delta_{FFT}^i$$

$$\therefore \Delta_{DP}^i \leq \Delta_{FFT}^i$$

When LoRA with low rank $r$ is applied, the weight updates get projected to a subspace $\mathcal{S}$ of rank $r$. Let the projection be $P_r : \mathbb{R}^{d \times k} \to \mathcal{S}_r$. We can frame the influence of the datapoint $i$ w.r.t LoRA model as:

$$\Delta_{LoRA}^i = ||\mathcal{P}_r(\Delta W(D)) - \mathcal{P}_r(\Delta W(D'))||_F = ||\mathcal{P}_r(g_i)||_F \; ; \mathcal{P}_r \text{ being a linear projection operator.}$$

Now, $||\mathcal{P}_r(g_i)||_F \leq ||g_i||_F$ as projection $\mathcal{P}_r$ will drop any point from $g_i$ lying outside its subspace.

$$\therefore \Delta_{LoRA}^i \leq \Delta_{FFT}^i \qquad \square$$

We can see that both LoRA and DP-SGD reduce the influence of the datapoint ; while DP-SGD does it probabilistically using noise $\sigma$, LoRA does it deterministically through low-rank projection $\mathcal{P}_r$. This aligns with prior work (Chaudhuri et al., 2013; Kapralov & Talwar, 2013) linking principal component analysis (PCA) or low-rank factorization to DP, where PCA limits leakage by discarding components (similar as in LoRA) and DP-SGD by adding noise.

## 7 CONCLUDING DISCUSSION

We investigated the interplay of privacy, utility, and efficiency in fine-tuning LLMs introducing measures that distinguish between sensitive and non-sensitive tokens. The conventional wisdom is that achieving privacy, using methods like DP-SGD, comes at the cost of computational efficiency. In contrast, we demonstrate that efficient fine-tuning methods like LoRA, initially designed for efficiency, achieves privacy benefits without any computational overhead. Simultaneously, LoRA retains the utility of language understanding compared to DP-SGD, or even FFT, highlighting its superiority in balancing all three objectives. Our paper calls for a joint venture of privacy and systems communities in achieving privacy-aware efficient fine-tuning of LLMs while retaining utility.

ETHICS STATEMENT

This study utilized publicly available datasets (Shi et al., 2022; Holmes et al., 2024), some of which included identifiable information such as personal details. However, third-party organizations pre-processed and validated the datasets to ensure that no real individuals' data were present, thus mitigating potential privacy concerns.

This project received ethical clearance from the Ethical Review Board of the affiliated institution on October 21, 2024 (Approval No. 24-09-4), with no ethical concerns raised.

REPRODUCIBILITY STATEMENT

To ensure the reproducibility of the results in the paper, we will release the code, execution environment, and artifacts publicly upon acceptance. The datasets – CustomerSim (Shi et al., 2022) and SynBio (Holmes et al., 2024) are publicly available. The scripts for fine-tuning are included in supplementary material. All hyperparameters for reproducing the results in the paper are listed in Appendix E.

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

APPENDIX

**Overview of Appendices**

## A    LIMITATIONS

One of the limitations of our work lies in fine-tuning the models for unsupervised setup and not extending it to other supervised tasks like question-answering and so on. However we believe that our findings would hold in any task setup, given that the nature of these fine-tuning techniques would not change. We experimented with LoRA as one the most generic PEFT methods – however, testing out other PEFT methods would be an interesting extension of our work to explore privacy benefits extensively in the systems community.

We consider the sensitive data to be from PIIs defined in GDPR Article 4(1) (EU, 2016). However, sensitive data can also be context-dependent. While one piece of information may be sensitive for one, it may not be for the other,. However, we also acknowledge that while contextual integrity is an important task, it is still hard to formalize or implement as seen in (Shvartzshnaider & Duddu, 2025). Lastly, while this work provides an empirical measure of privacy across different fine-tuning methods, one can definitely use such a measure for optimisation during training and establish a theoretical bound on the privacy benefits that would then also be empirically validated. We intend to explore these directions in the future.

## B    USE OF LLMS FOR PAPER-WRITING

In this paper, we use LLMs for the following purposes:

1. **Polishing the writing**: We check for grammatical mistakes, and make minor para-phrasing to improve the flow and coherence of the paper.

2. **Related Work**: Besides traditional search, we use the OpenAI Deep Research to identify relevant literature.

## C    EXTENDED RELATED WORK

**Privacy attacks.** Privacy concerns in LLMs have gained attention in recent years Kibriya et al. (2024); Yao et al. (2024), particularly the possibility of exposing data via membership inference attacks (MIAs) Mireshghallah et al. (2022a); Mattern et al. (2023); Fu et al. (2023); Kaneko et al. (2024) and training data extraction attacks (DEAs) Carlini et al. (2021); Mireshghallah et al. (2022b); Panda et al. (2024). While MIAs have the goal of identifying if a certain datapoint was in the training dataset using model confidence scores, DEAs try to extract specific parts of data from the training dataset using different prompting strategies. A recent study Zhang et al. (2024) underscores that membership inference attacks may not be reliable due to their practical limitations of requiring knowledge of the entire training data. In this work, we focus on whether different fine-tuning methods mitigate the privacy risks for DEAs as they are more practical and effective mode of attacksZhang et al. (2024).

**Privacy mitigation.** To mitigate privacy leakage through MIAs, differential privacy (DP) measures have been proposed, which add theoretical privacy guarantees to the training process Abadi et al. (2016) with no empirical validation. The key idea of DP is to reduce the influence of datapoints during training and reduce the chances of leakage during inference with some formal guarantees. However, these DP guarantees have been shown to not strongly hold for data reconstruction attacks in Hayes et al. (2023). In this work, we measure the empirical privacy benefits of DP on DEAs and find it to be effective.

**Privacy measurement.**The use of DP in LLMs has been criticized for offering privacy guarantees that are insufficient unless sensitive data is removed Brown et al. (2022); Tramèr et al. (2024). This finding raises the question of whether current measures of privacy is adequate. Intuitively, privacy evaluation should focus on the sensitive data encountered during training. While some studies on LLM privacy attacks consider sensitive vs. non-sensitive data during training Kim et al. (2024); Lukas et al. (2023); Zhao et al. (2022); Shi et al. (2022), they often ignore this distinction in evaluation. In our work, we preserve this distinction and propose a revised quantification that holds for DEAs.

Table 1: Example of samples from datasets with sensitive tokens highlighted.

| CustomerSim | |
|---|---|
| SYS: Hello, I am the customer support bot. What can I do for you? USR: Hello robot. I ordered a pot several days ago but I can't track it. SYS: Could you verify your full name? USR: Patrick Schug SYS: Verify your order number please. USR: It's 843-58572-7002. SYS: You can track your package with your tracking number , which is AGZIM5T6KL. Are you happy about my answer? USR: All good. See you. SYS: Have a nice day! Bye. | SYS: Hello, I am the customer support bot. What can I do for you? USR: Hi. Where is my package? SYS: Could you verify your full name? USR: I am Catherine Pena. SYS: Verify your phone number. USR: You can reach me at 547.302.3744. SYS: The tracking number is VVTPHDB6VK. Anything else? USR: All good. |
| **SynBio** | |
| My name is Alexander Tanaka, and I'm a saleswoman with a year of experience. I recently completed a project that involved developing and implementing a new sales strategy for my company. I started by analyzing our current sales data to identify areas where we could improve... | My name is Phillip Martinez, and I would like to share some aspects of my life's journey with you. I have had the pleasure of living in various places throughout my life, but I currently reside at 4537 Tanglewood Trail ... you can reach me via email at phillip-martinez@outlook.com or by phone at +86 19144 1648. |

**Privacy-Efficiency tradeoff.** Additionally, to ensure privacy guarantees, DP requires higher computational resources (time and memory) during training. For example, compared to vanilla SGD, DP-SGD may incur up to 20x training time, which is often a bottleneck in resource-intensive tasks Dupuy et al. (2022). Therefore, the conventional wisdom is that *privacy comes at the cost of efficiency*. In this work, we probe ways in which the above limitation can be prevented.

**Efficiency-Utility tradeoffs.** Full fine-tuning (FFT) of large language models is expensive as it involves updating all the parameters. Recent work has focused on reducing the training cost while maintaining utility, with the introduction of parameter efficient fine-tuning (PEFT) techniques Han et al. (2024). Well-known PEFT methods include adapter-based fine-tuning Houlsby et al. (2019); Lei et al. (2023); Zhu et al. (2021); He et al. (2021), prefix-tuning Li & Liang (2021a), soft prompt-based fine-tuning Li & Liang (2021b); Li et al. (2023); Liu et al. (2021; 2024b); Lester et al. (2021), P-tuning Liu et al. (2022), and parameterized fine-tuning Hu et al. (2022); Liu et al. (2024a). Among different PEFT methods, we focus on LoRA Hu et al. (2022) as it is more generic, while the rest are task-specific and mostly limited to supervised settings.

**Privacy-Efficiency-Utility tradeoffs.** In an ideal world, the best scenario would be to achieve all the three objectives. While DP is shown to achieve privacy and utility, compromising on efficiency, LoRA is shown to achieve utility and efficiency at an unknown cost of privacy, that we try to characterize in this work. We observe that LoRA achieves similar level of privacy as DP. In recent work on private efficient fine-tuning Liu et al. (2023); Yu et al. (2021); Ma et al. (2024), the authors combine LoRA with DP to reduce the additional computational overhead induced by DP. However, in our work, we show that one can directly achieve all the three objectives — privacy, utility, and efficiency — by only using LoRA with no additional overhead.

## D  DATASET SAMPLE

Table 1 lists two samples from each of the two datasets – Customersim (Shi et al., 2022) and SynBio (Holmes et al., 2024).

## E  HYPERPARAMETERS

All experiments were conducted on a SLURM cluster consisting of nodes equipped with NVIDIA A100 and H100 GPUs. During fine-tuning, we use `Trainer` module from Transformers Wolf et al. (2020), and specifically `peft` library Mangrulkar et al. (2022) for LoRA and the `opacus` library Yousefpour et al. (2021) for DP. Table 2 and 3 lists the hyperparameters used for fine-tuning experiments. We report our results on a single run with a fixed seed of $42$.

| Hyperparameter | Full Fine-Tuning (FFT) | LoRA | DP |
|---|---|---|---|
| Learning Rate | 0.00025 | 0.00025 | 0.00005 |
| Scheduler | Linear | Linear | Linear |
| Warmup Steps | 10 | 10 | 10 |
| Rank ($r$) | - | $\{16, 32\}$ | - |
| Scaling factor ($\alpha$) | - | $\{16, 32, 64, 128\}$ | - |
| Noise ratio ($\sigma$) | - | $\{0.1, 0.5, 0.9\}$ | - |
| Clipping gradient threshold ($T$) | - | - | $(1 \times 10^{-2})$ |

Table 2: Hyperparameters used for fine-tuning methods

| Dataset | Model | Full Fine-Tuning (FFT) | LoRA | DP |
|---|---|---|---|---|
| CustomerSim | Pythia | 16 | 32 | 8 |
| | Gemma | 8 | 16 | 4 |
| | Llama2 | 4 | 8 | 2 |
| | Qwen2.5 | 4 | 8 | 2 |
| SynBio | Pythia | 8 | 16 | 4 |
| | Gemma | 4 | 8 | 2 |
| | Llama2 | 2 | 4 | 2 |
| | Qwen2.5 | 2 | 4 | 2 |

Table 3: Batch sizes for CustomerSim and SynBio

## F    PROMPT FOR ANNOTATING PRIVACY-SENSITIVE INFORMATION

We used the following prompt to obtain the privacy-sensitive information from GPT-4o for CustomerSim:

*"You are a private data identifying bot. You will be provided a conversation between a user and a customer service bot. You need to identify ALL instances of private data in the conversation. Private data includes any information that can be used to identify an individual, such as names, phone numbers, email addresses, and locations. It can also include any other sensitive information, such as credit card numbers, social security numbers, tracking data, health information etc.. Identify all of these use your discretion to identify any other information that can be used to identify an individual."*

## G    HUMAN SURVEY: ASSESSING THE QUALITY OF PRIVACY-SENSITIVE ANNOTATIONS FROM PRESIDIO AND GPT-4

We evaluated privacy-sensitive annotations from Presidio and GPT-4 through a Prolific survey, each with 10 samples for 40 participants. Each of the 10 instances included 5 samples from each tool, displayed randomly. Participants had four options: (1) *Accurate* (all sensitive info correctly annotated), (2) *Under-Annotated* (some sensitive info missed), (3) *Over-Annotated* (non-sensitive info included), and (4) *Mixed-Annotated* (both missed and non-sensitive info annotated). Figure 8 shows sample screenshots of the conducted survey. Figures 9a and 9b shows GPT-4 annotations as more "Accurate" than Presidio.

## H    WHY DISTINGUISH BETWEEN SENSITIVE AND NON-SENSITIVE ENTITIES?

Figure 10 shows the assessment of privacy and utility by distinguishing between sensitive and non-sensitive tokens for all models. We also show the utility measurements in log-scale plots in Figure 11 for better understanding.

## I    EXPLORING THE TRADEOFFS BETWEEN PRIVACY AND UTILITY

**Full Fine-tuning**: We see a similar trade-off between utility and privacy when using Presidio annotations in Figure 12, as observed in Figure 3. For the CustomerSim dataset, we notice that privacy decreases at the beginning of training, while utility improves. However, as training progresses, both metrics start to worsen. In contrast, the SynBio dataset shows a decline in both utility and privacy,

(a) Instructions for annotators

(b) Guidelines for annotators

(c) Sample Question 1

(d) Sample Question 2

Figure 8: Annotators were provided with these instructions and guidelines to mark the samples as accurately annotated, under-annotated, over-annotated or mixed-annotated.

except for the Pythia model, which initially experiences an improvement in utility during the first few epochs.

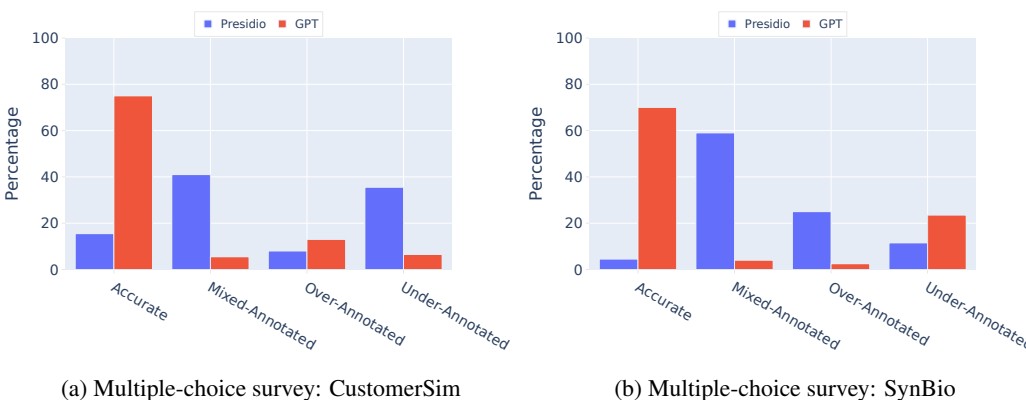

(a) Multiple-choice survey: CustomerSim          (b) Multiple-choice survey: SynBio

Figure 9: GPT-4 was preferred in both CustomerSim and SynBio by human annotators for identifying privacy-sensitive information.

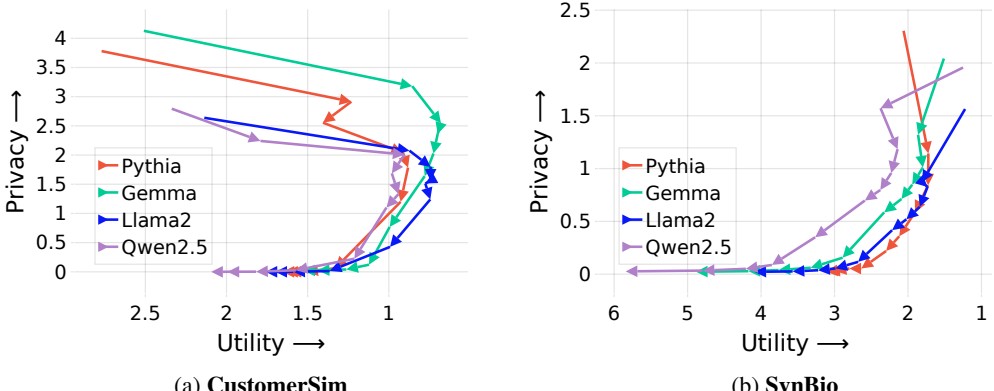

(a) **CustomerSim**          (b) **SynBio**

Figure 12: **Full fine-tuning** on (a) CustomerSim and (b) SynBio using Presidio to annotate the privacy-sensitive information.

**DP-SGD** : A similar pattern as in Figure 4 using GPT-4 annotations, has also emerged in Figure 13 using Presidio annotations. We observe that in the CustomerSim dataset, DP maintains privacy effectively with minimal degradation. However, when it comes to utility, we notice that lower noise values lead to better utility, but also result in a decrease in privacy. The same holds true for the SynBio dataset in Figure 14, with the only exception being Gemma experiencing a decline in utility after a few iterations.

**LoRA** : Comparing the results from Figures 15 and 16 using Presidio annotations with Figure 5, which used GPT-4 annotations, we can observe the same trend. This suggests that the findings are consistent across different annotation methods. Similarly, when we analyze the trade-off for rank 32 for both the datasets in Figures 17 - 20, as we observed in Figure 5 using GPT-4 annotations.

We analyze the trade-offs between privacy, utility, and efficiency for different privacy measures over SynBio with both GPT4 and Presidio annotations. Figures 21 , 22, 23 for CustomerSim and SynBio using Presidio annotations and our privacy measure reveals the same as observed in Figure 6 with GPT-4 annotations. We also show the same for the other privacy measures – (a) privacy loss in Figures 24, 25, and 26 , and (b) exposure in Figure 27.

The overall takeaway holds consistent – LoRA is the most efficient method, followed by FFT while DP is the least efficient, requiring the highest number of FLOPs. For privacy, DP and LoRA perform

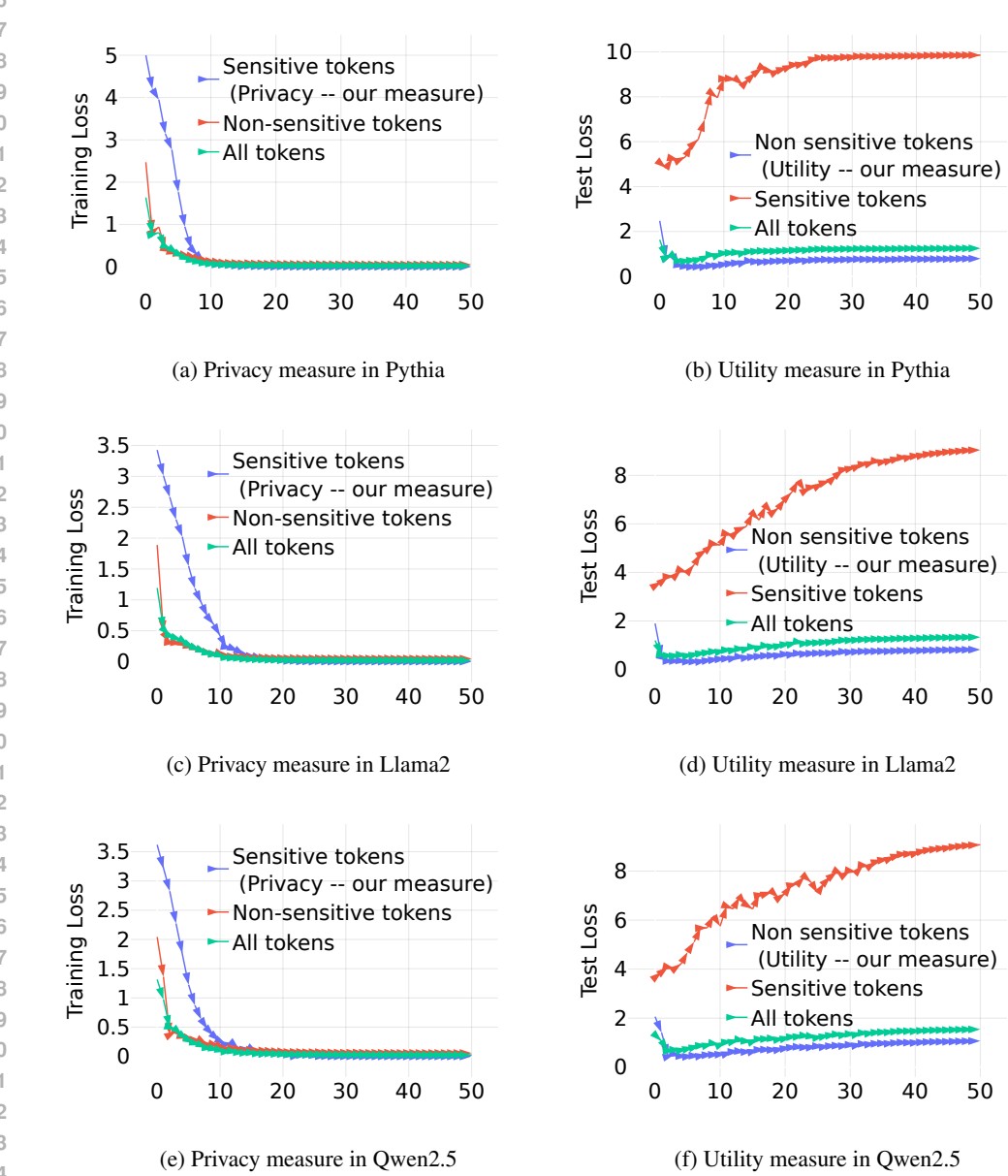

(a) Privacy measure in Pythia

(b) Utility measure in Pythia

(c) Privacy measure in Llama2

(d) Utility measure in Llama2

(e) Privacy measure in Qwen2.5

(f) Utility measure in Qwen2.5

Figure 10: Our measures offer a precise assessment of privacy and utility by distinguishing between sensitive and non-sensitive tokens.

at par with FFT leaking the most amount of information. In terms of utility, LoRA and FFT perform better while DP deteriorates on large scale models. Additionally, LoRA also maintains benchmark performance across other models as shown in Figure 28.

## J COMPARISON OF FINE-TUNING METHODS

Based on our results, we provide a taxonomy on the axes of privacy, utility, and efficiency for all the three fine-tuning methods in Table 4.

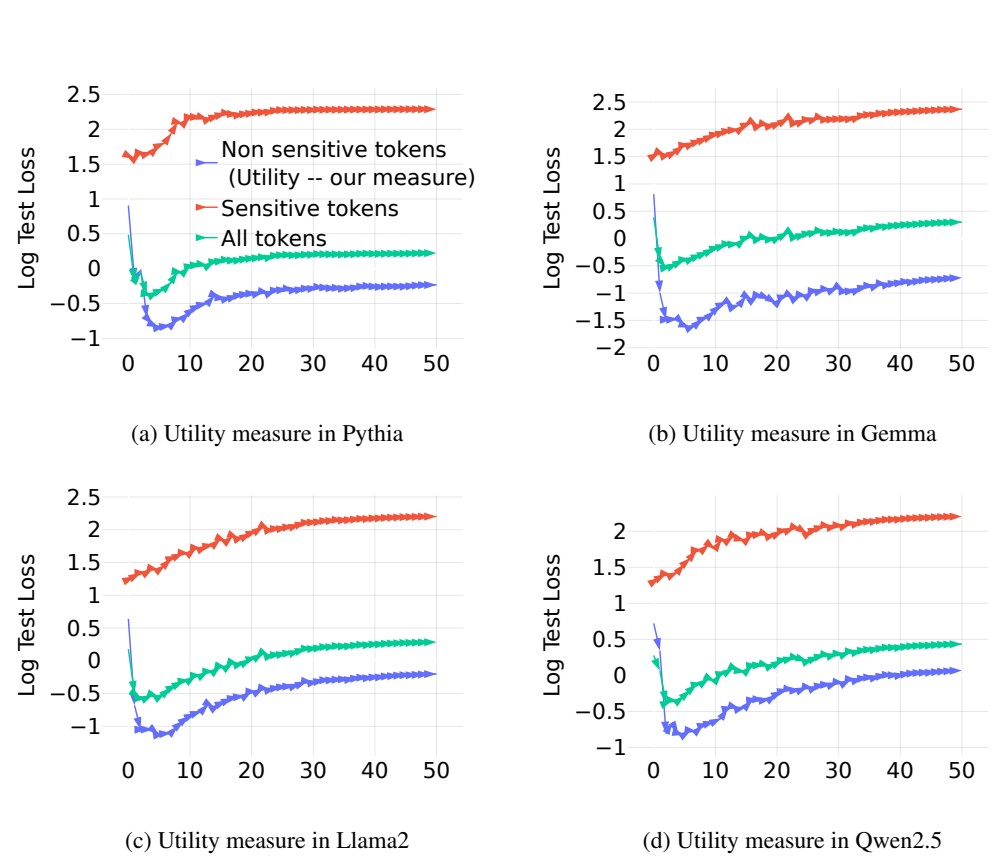

(a) Utility measure in Pythia

(b) Utility measure in Gemma

(c) Utility measure in Llama2

(d) Utility measure in Qwen2.5

Figure 11: Our measures offer a more precise assessment of utility when fine-tuning LLMs by distinguishing between sensitive and non-sensitive tokens, revealing better utility (lower loss) for non-sensitive tokens compared to scenarios that overlook the sensitivity-based distinction.

| | Utility | Privacy | Efficiency | Benchmark |
|---|---|---|---|---|
| FFT | Good | Poor | Moderate | Poor |
| DP | Moderate | Good | Poor | Poor |
| LoRA | Good | Good | Good | Good |

Table 4: Comparison of all fine-tuning methods across privacy, utility, efficiency, and benchmark performance. LoRA is seen to outperform FFT and DP over all dimensions.

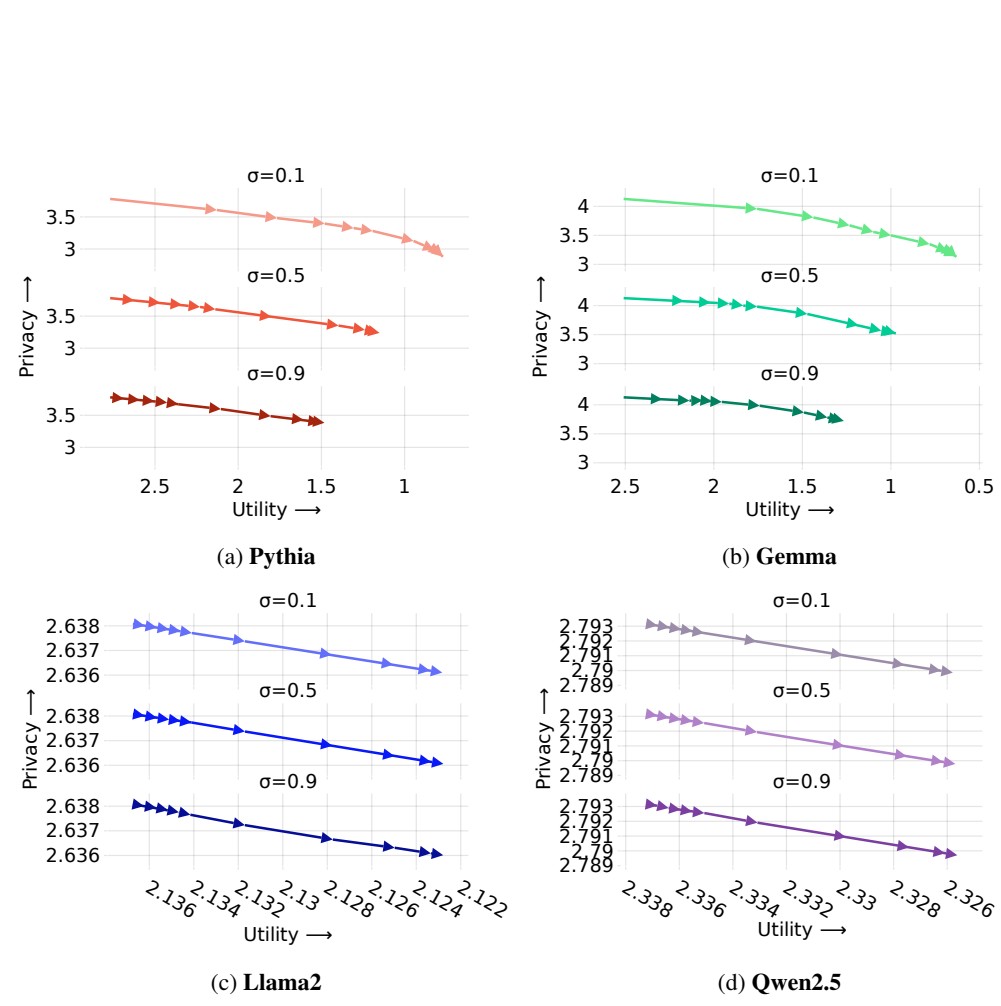

Figure 13: **DP** on *CustomerSim* dataset using Presidio tool for annotating privacy-sensitive information.

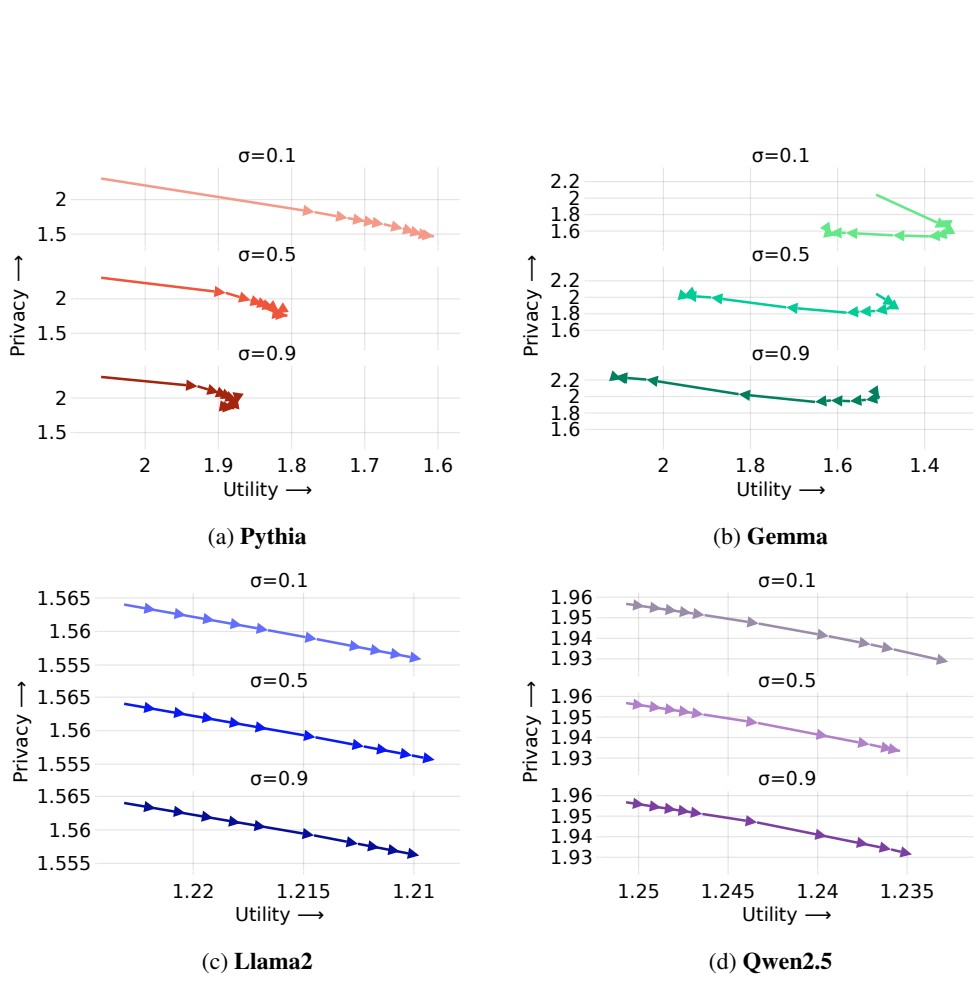

(a) **Pythia**

(b) **Gemma**

(c) **Llama2**

(d) **Qwen2.5**

Figure 14: **DP** on *SynBio* dataset using Presidio tool for annotating privacy-sensitive information.

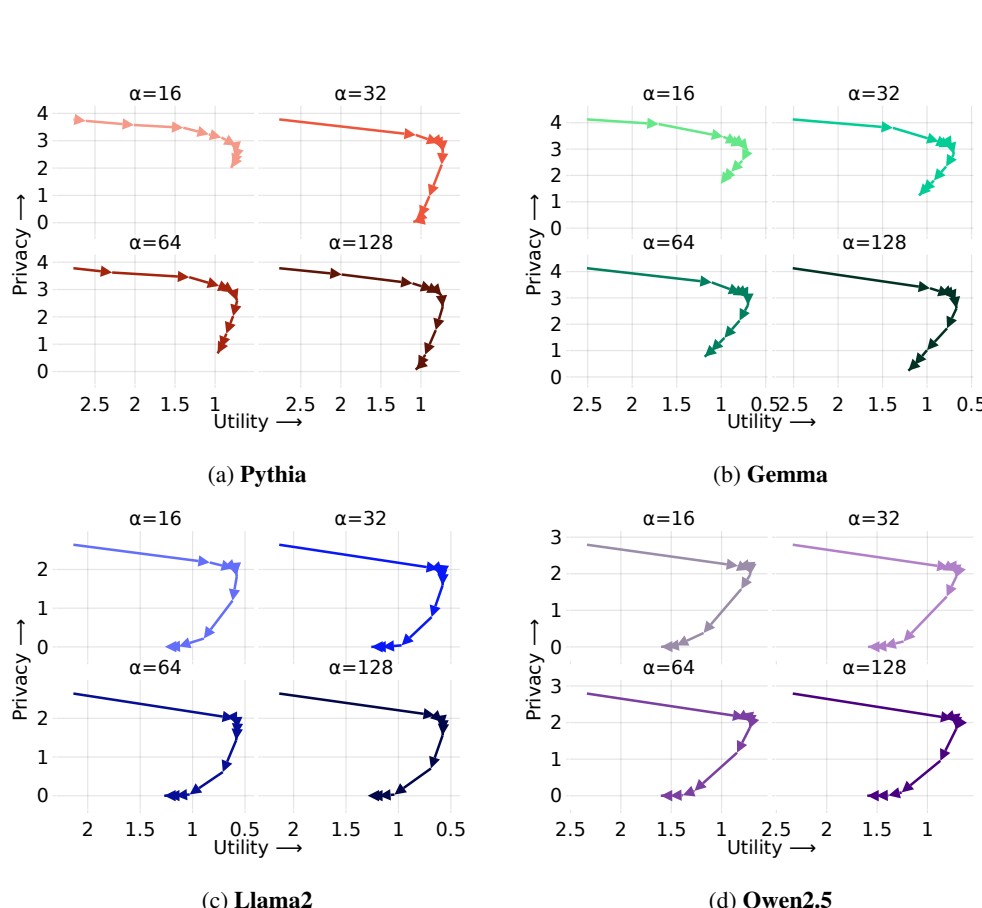

(a) **Pythia**

(b) **Gemma**

(c) **Llama2**

(d) **Qwen2.5**

Figure 15: **Lora with rank 16** on *CustomerSim* dataset using Presidio tool for annotating privacy-sensitive information.

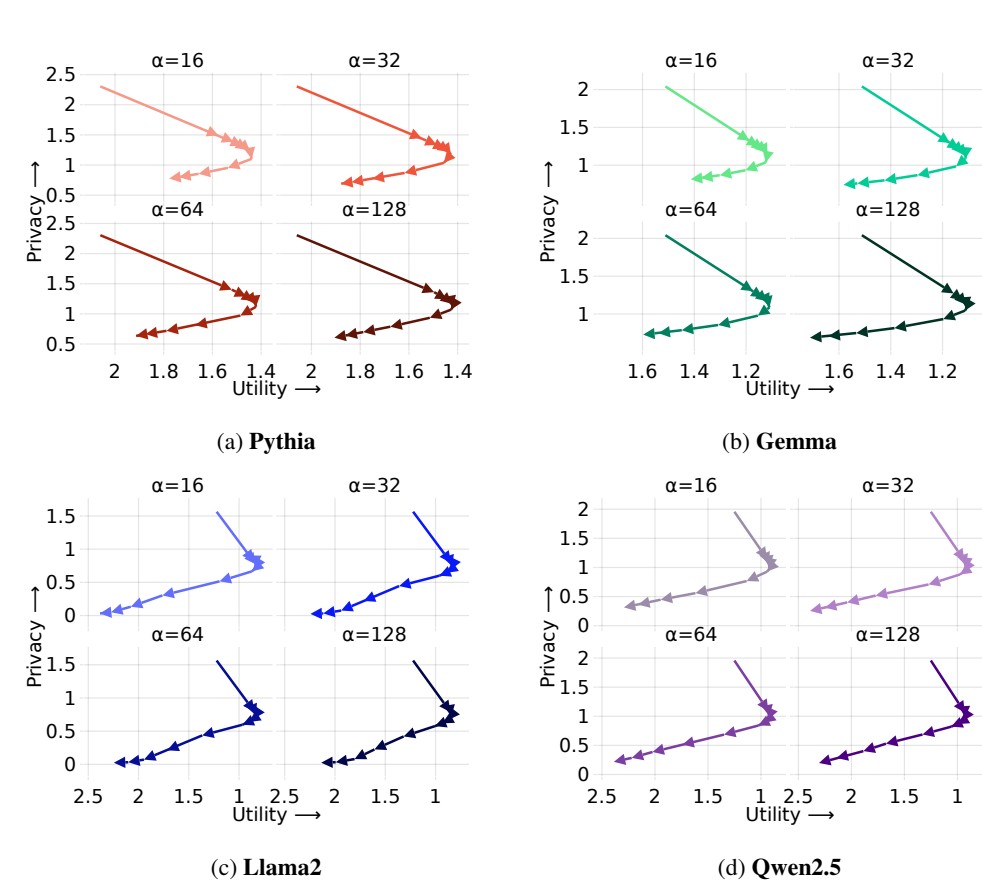

(a) **Pythia**

(b) **Gemma**

(c) **Llama2**

(d) **Qwen2.5**

Figure 16: **Lora with rank 16** on *SynBio* dataset using Presidio tool for annotating privacy-sensitive information.

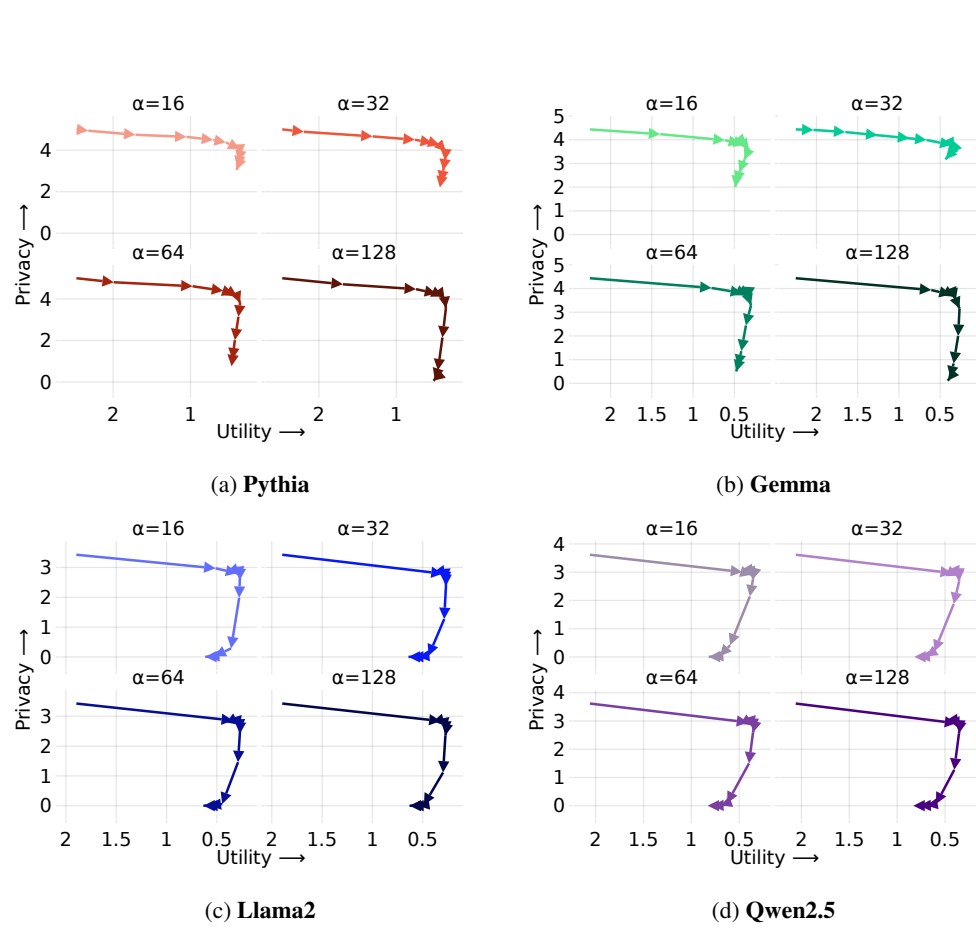

Figure 17: **Lora with rank 32** on *CustomerSim* dataset using GPT4 for annotating privacy-sensitive information.

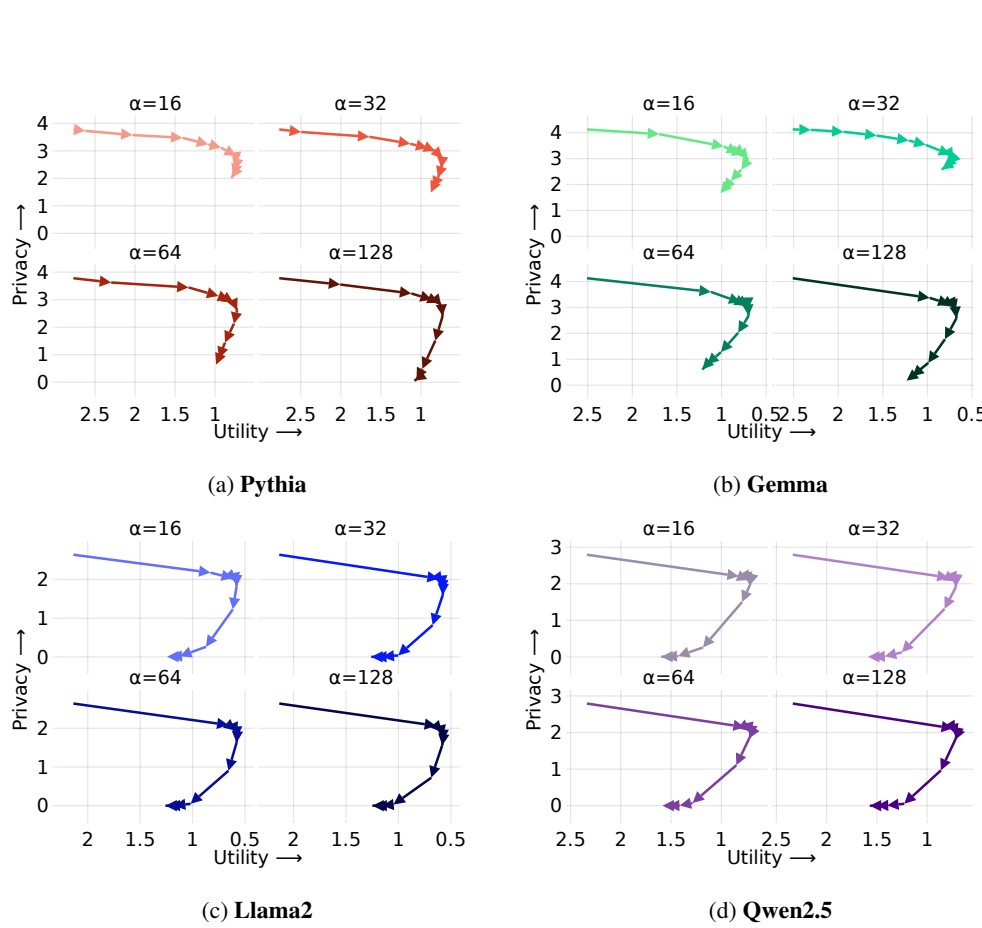

(a) **Pythia**

(b) **Gemma**

(c) **Llama2**

(d) **Qwen2.5**

Figure 18: **Lora with rank 32** on *CustomerSim* dataset using Presidio tool for annotating privacy-sensitive information.

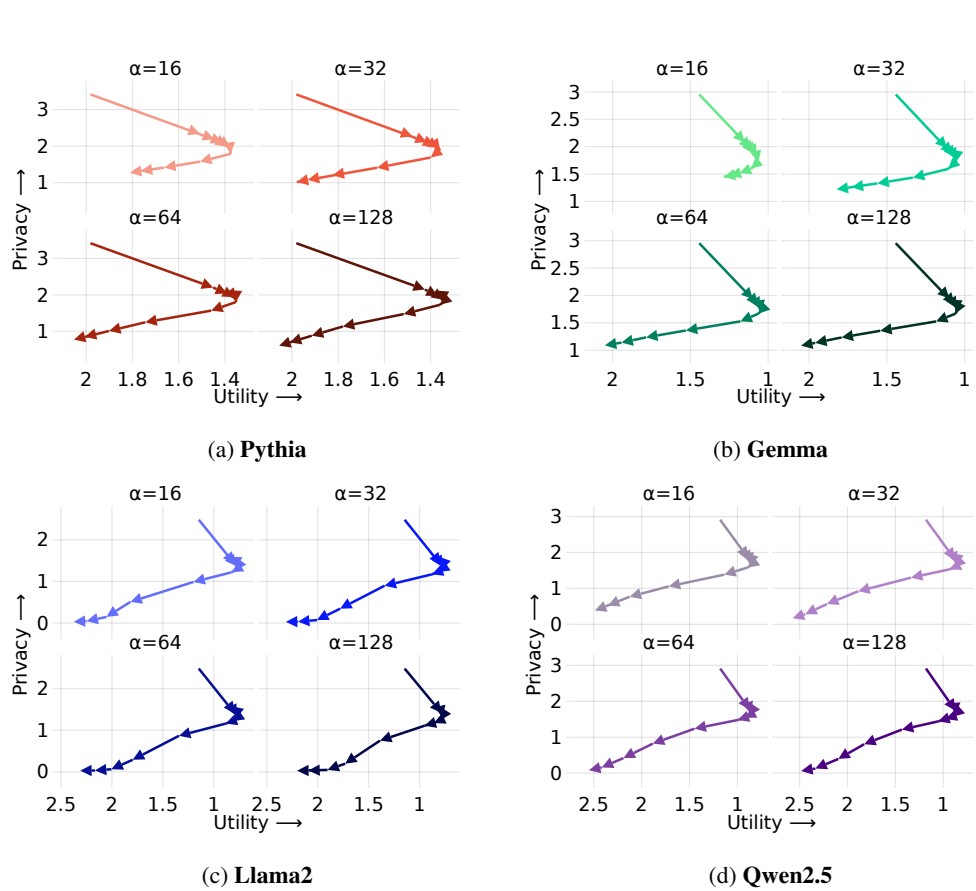

Figure 19: **Lora with rank 32** on *SynBio* dataset using GPT4 for annotating privacy-sensitive information.

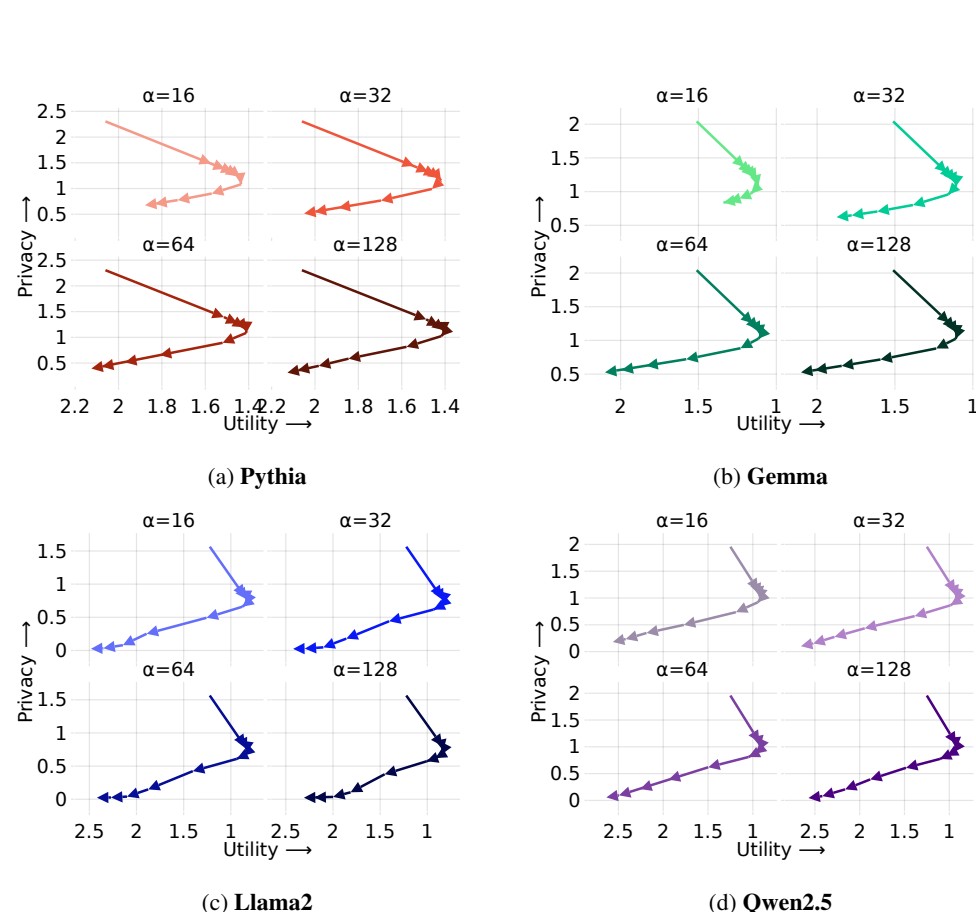

(a) **Pythia**

(b) **Gemma**

(c) **Llama2**

(d) **Qwen2.5**

Figure 20: **Lora with rank 32** on *SynBio* dataset using Presidio tool for annotating privacy-sensitive information.

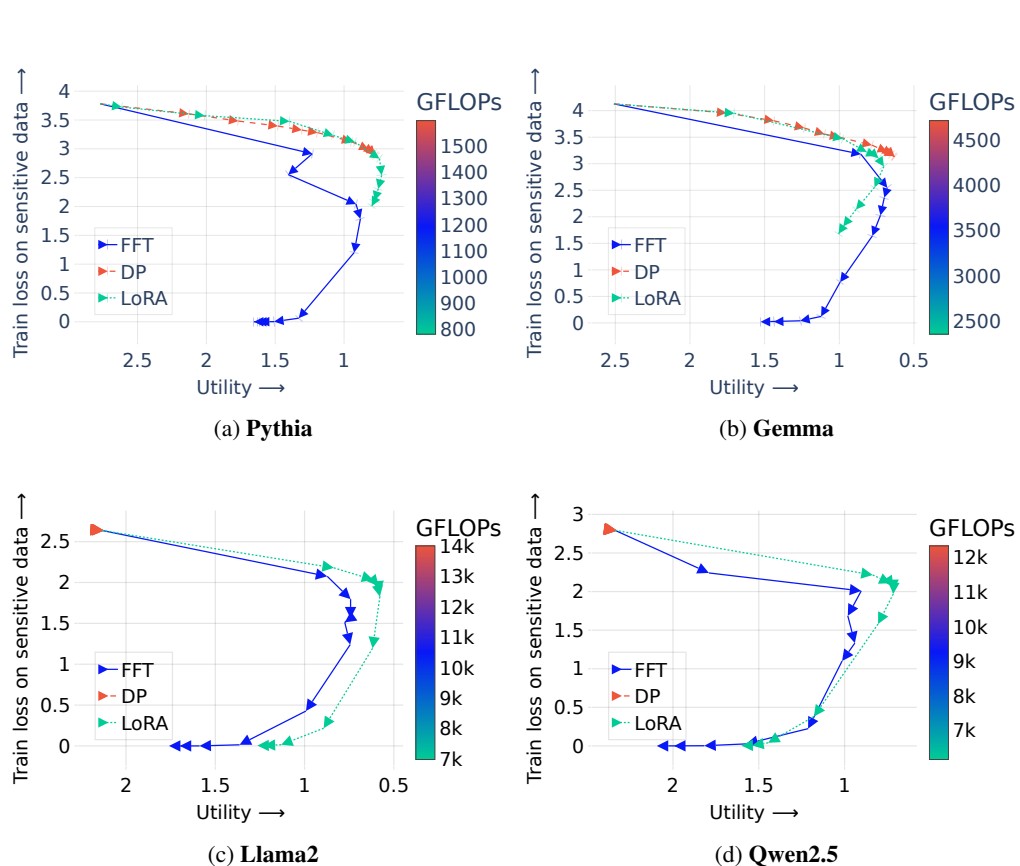

Figure 21: **Full fine-tuning, DP and LoRA** on *CustomerSim* with Presidio annotations for privacy-sensitive information.

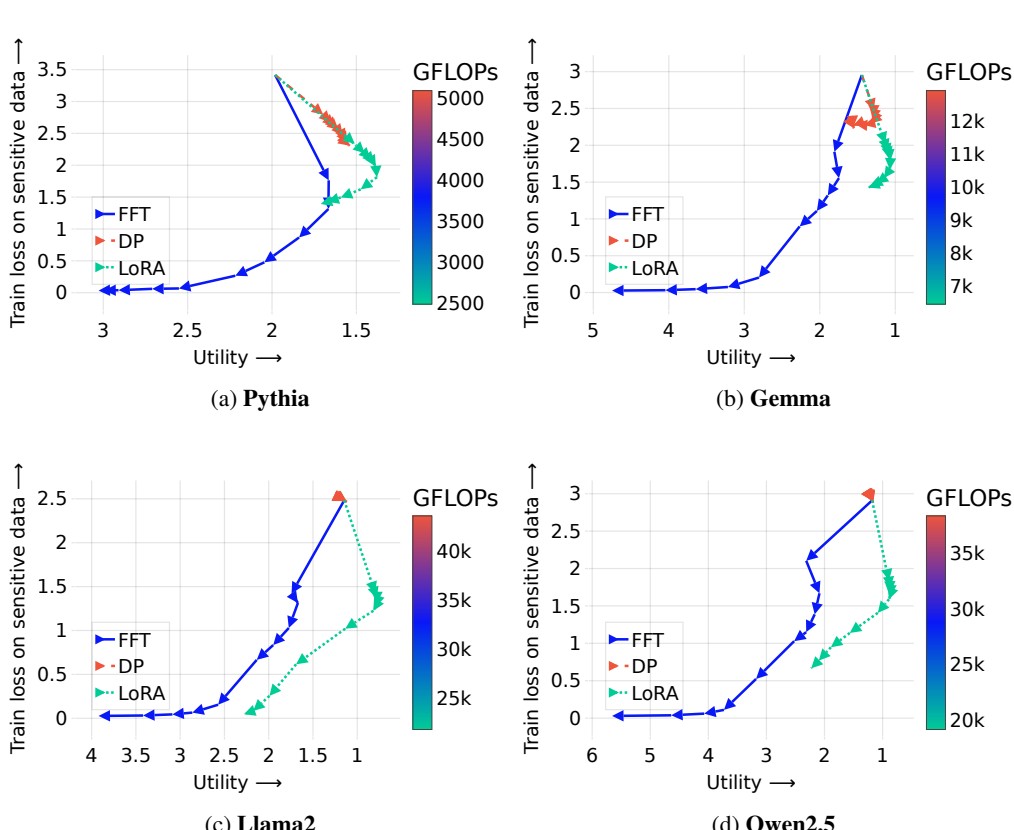

(a) **Pythia**

(b) **Gemma**

(c) **Llama2**

(d) **Qwen2.5**

Figure 22: **Full fine-tuning, DP and LoRA** on *SynBio* with GPT4 annotations for privacy-sensitive information.

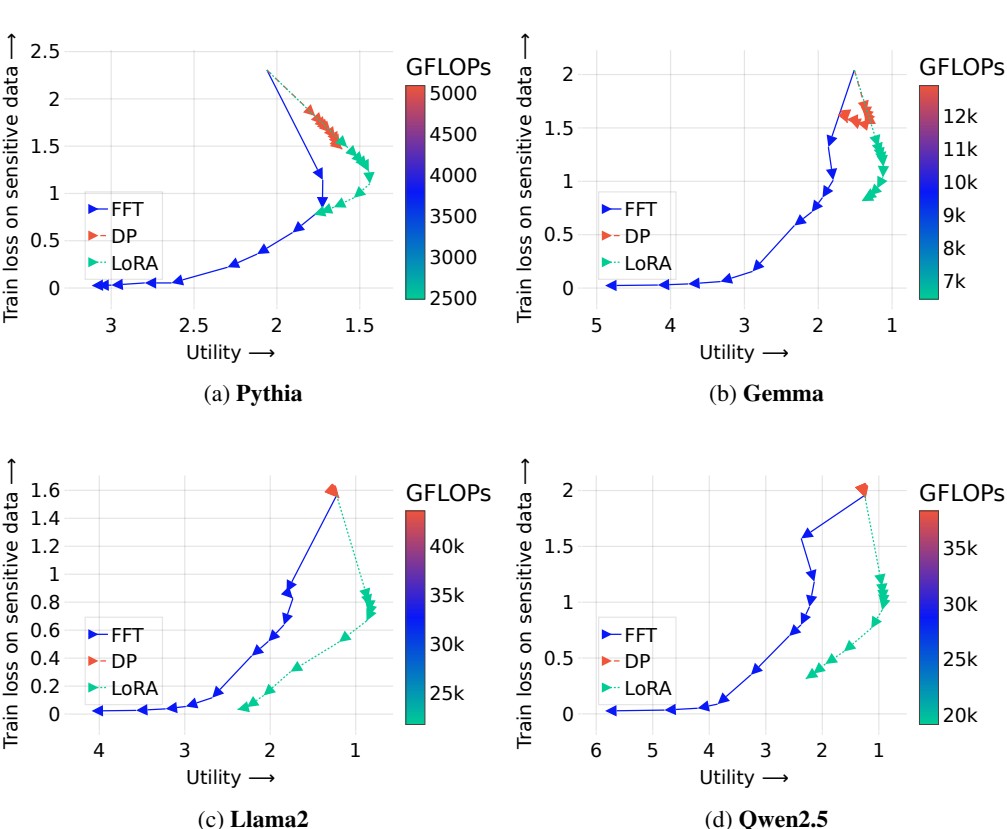

Figure 23: **Full fine-tuning, DP and LoRA** on *SynBio* with Presidio annotations for privacy-sensitive information.

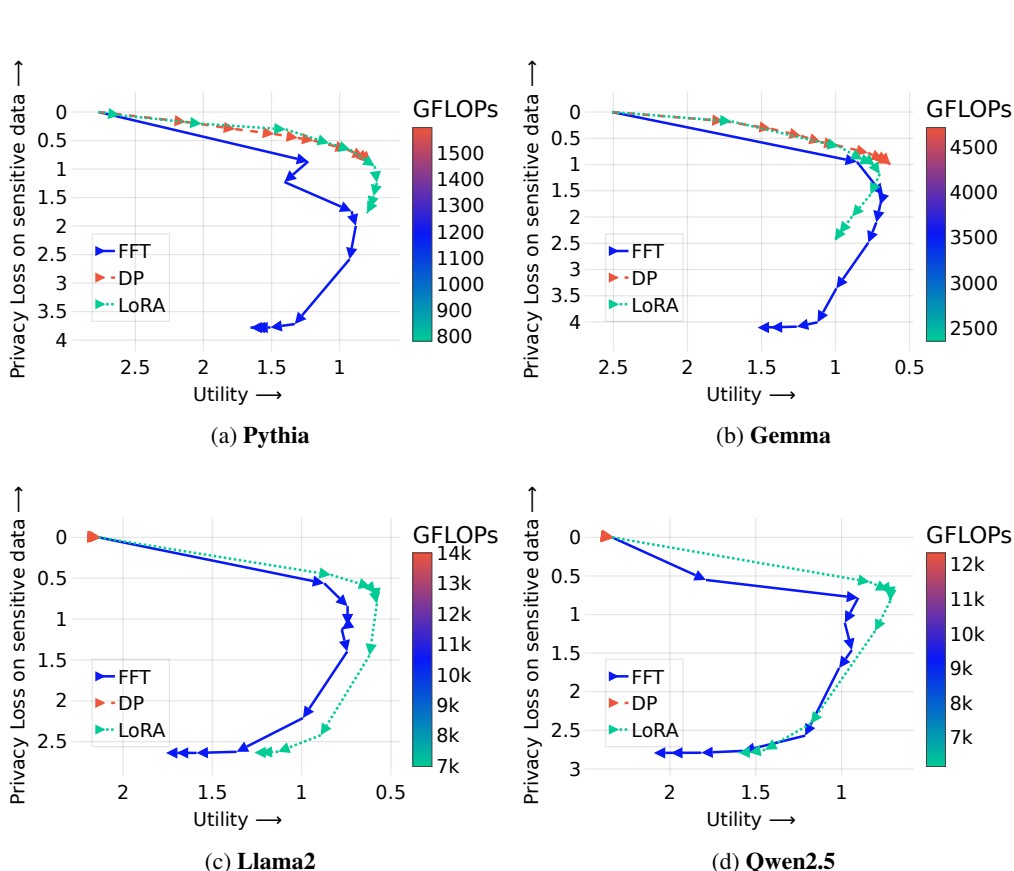

Figure 24: **Full fine-tuning, DP and LoRA** on *CustomerSim* with Presidio annotations for privacy loss

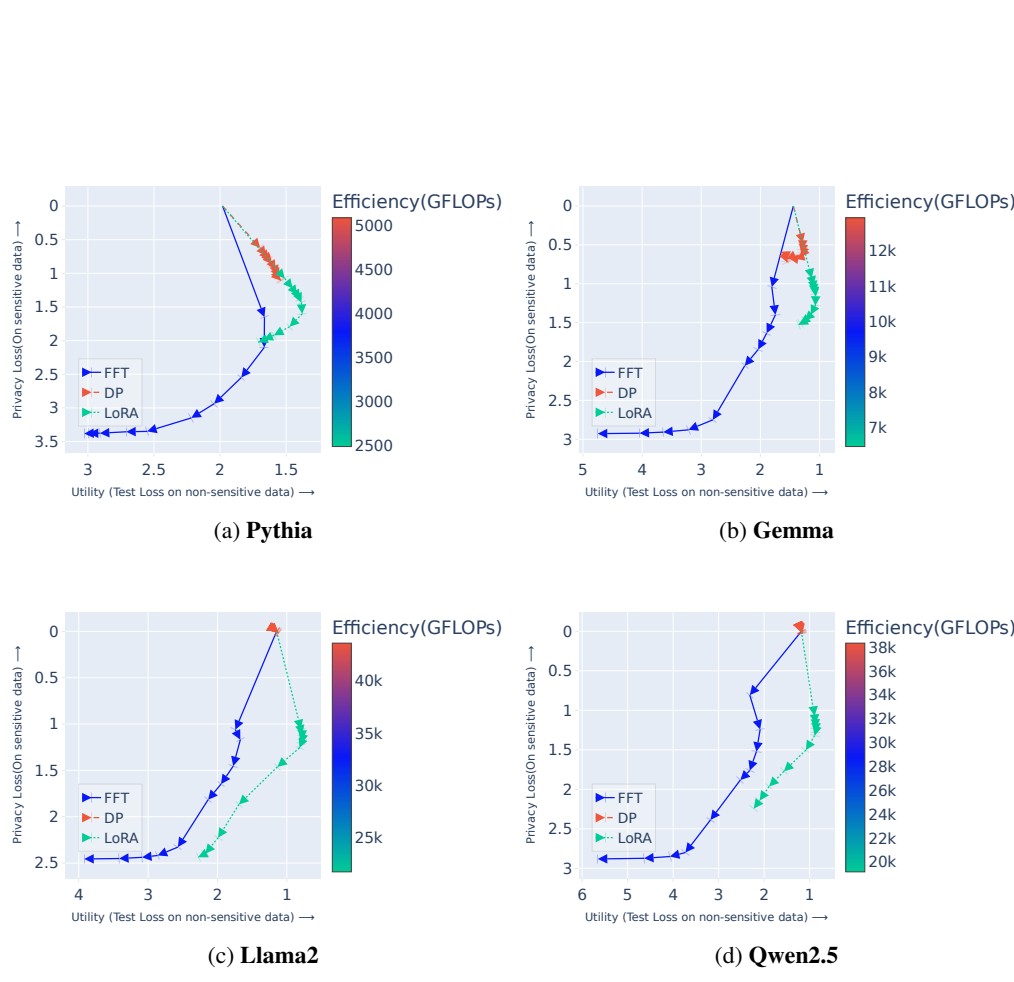

Figure 25: **Full fine-tuning, DP and LoRA** on *SynBio* with GPT4 annotations for privacy loss

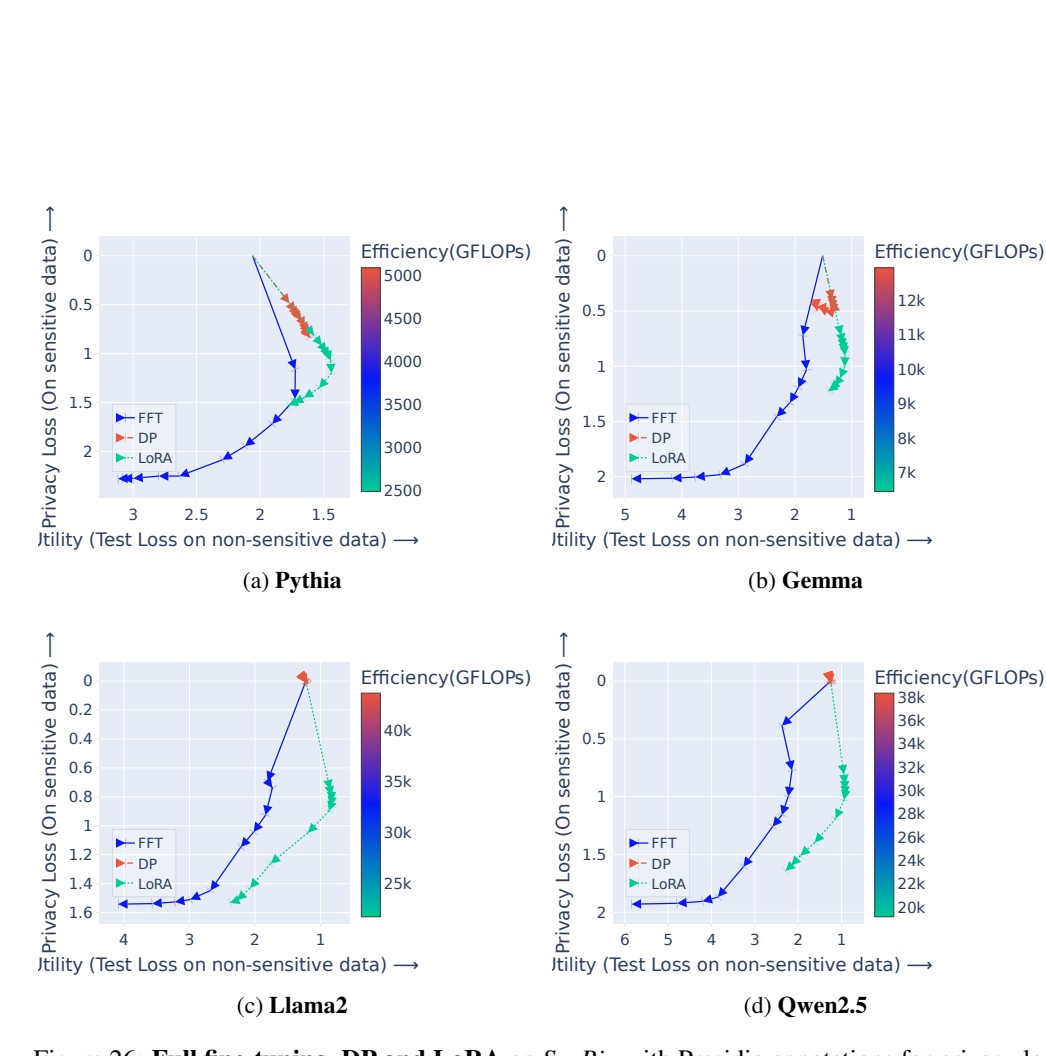

Figure 26: **Full fine-tuning, DP and LoRA** on *SynBio* with Presidio annotations for privacy loss

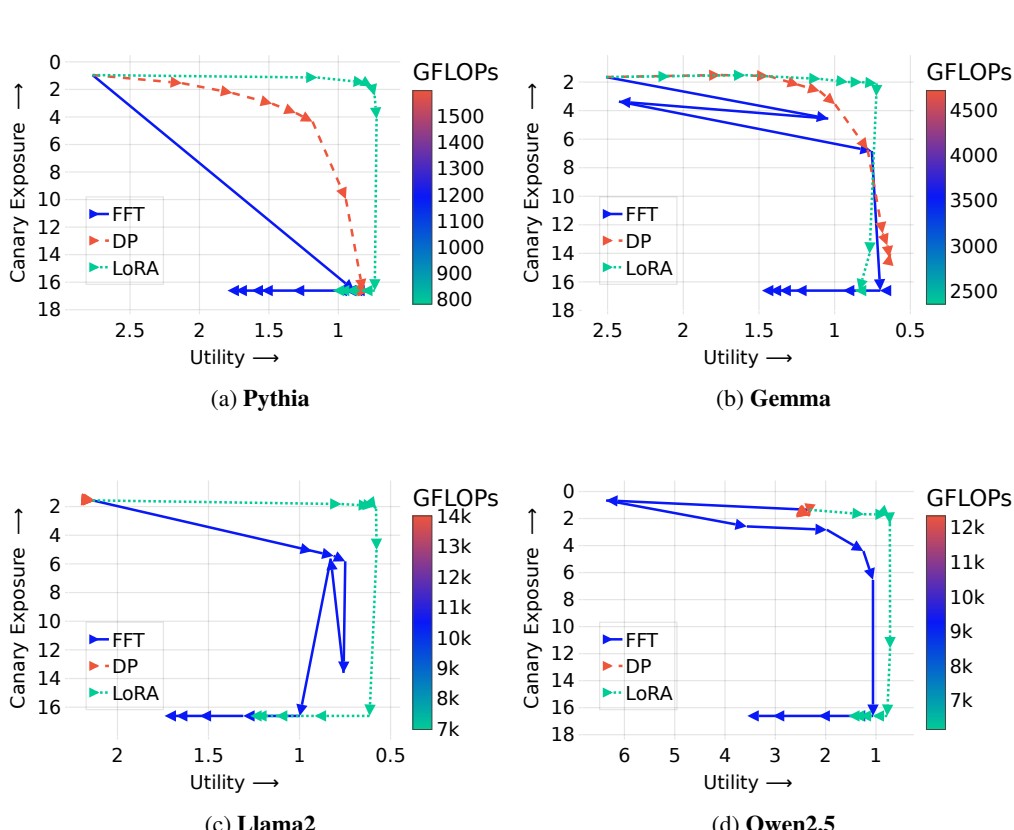

Figure 27: **Full fine-tuning, DP and LoRA** on *CustomerSim* with Presidio annotations for canary exposure.

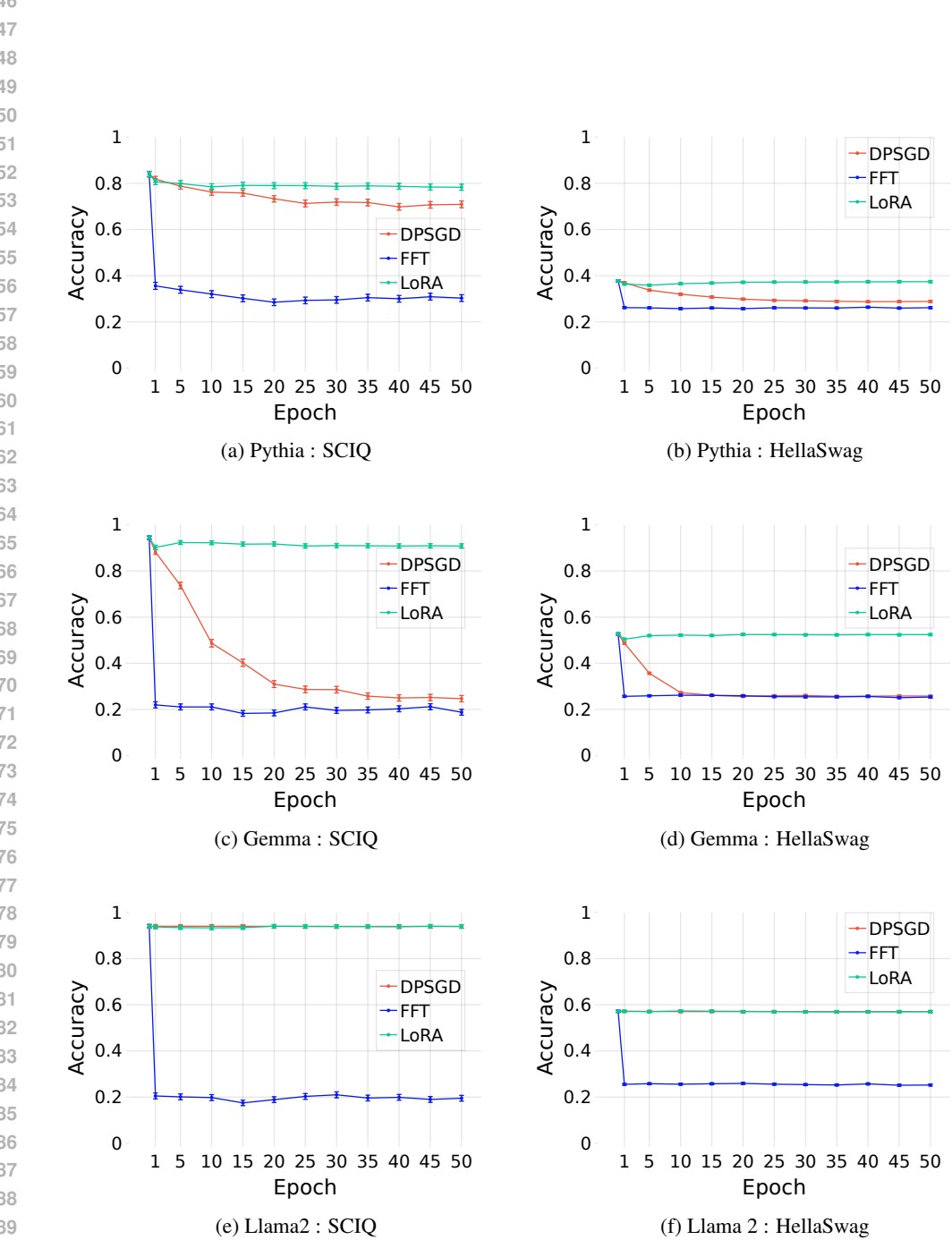

Figure 28: LoRA demonstrates knowledge retention on benchmark datasets throughout the training regime on CustomerSim for other models as well, while FFT and DP show fragility with sharp and gradual performance declines, respectively.

