# OpenReview forum: "Revisiting Privacy, Utility, and Efficiency Trade-offs when Fine-Tuning Large Language Models"
_ICLR.cc/2026/Conference — Submitted to ICLR 2026_

### Official Review · Reviewer_SuFS · 2025-10-25

**Soundness:** 2
**Presentation:** 3
**Contribution:** 2
**Rating:** 2
**Confidence:** 4

**Summary:**

This paper studies the privacy-utility-efficiency tradeoffs when fine-tuning LLMs with full fine-tuning (FFT), DP-SGD, and LoRA. They measure privacy as the model’s ability to recollect sensitive tokens in the training data, and utility as the model’s ability to predict non-sensitive tokens in the test data. They run a systematic study using Pythia, Gemma, Llama and Qwen models, using two datasets. Overall, they find that FFT gives poor utility-privacy tradeoffs, DP-SGD has reasonable utility-privacy tradeoffs but is computationally expensive, and LoRA almost achieves similar privacy to DP-SGD while being more computationally efficient.

**Strengths:**

- The paper shows empirical evidence that LoRA might preserve privacy while being computationally efficient.
- The authors systematically study the tradeoffs between privacy, utility and efficiency for the three different fine-tuning methods, on a range of models.

**Weaknesses:**

- The paper only presents some intuitive arguments and some experimental evidence to show that LoRA preserves privacy, but there are no theoretical guarantees. Given that the authors use a nonstandard privacy measure, it would also be good to compare against, for example, membership inference attacks to validate the empirical privacy claims.
- The GPT-4 annotation quality for the sensitive/non-sensitive tokens is questionable, given how only 75% of the Prolific participants found the GPT-4 annotations to be accurate. It would be good to provide more details about the annotation results, for instance the false positive/false negative rates compared to GPT-4, and the agreement/disagreement rate between the participants. It would also be good to include a discussion on how much the results of this paper are affected if the annotations are only 75% correct.
- The datasets used are simulated or synthetic, and may not be reflective of real-world datasets related to privacy.
- The measure of utility is non-standard, and it would be good to also compare against more standard utility measures that could be more task-specific.

**Questions:**

- How would misclassifications of sensitive/non-sensitive tokens by using GPT-4 affect the results in the paper?
- How do the privacy/utility measures used in this paper compare against more standard measures used in the literature?

---

### Official Review · Reviewer_j2Cg · 2025-10-31

**Soundness:** 2
**Presentation:** 3
**Contribution:** 3
**Rating:** 4
**Confidence:** 4

**Summary:**

The paper examines how different fine-tuning methods impact privacy, computational efficiency, and model performance. It introduces new measures to distinguish between sensitive and non-sensitive token recollection, showing that these distinctions are crucial for evaluating privacy and utility. Through experiments on models like Pythia and Qwen, the authors find that Low-Rank Adaptation achieves privacy levels comparable to Differential Privacy while being far more computationally efficient. The work challenges the traditional belief that improving privacy must come at the cost of efficiency, demonstrating that LoRA can balance privacy utility and efficiency simultaneously

**Strengths:**

1. Broad empirical study across multiple model families datasets and fine tuning methods with consistent comparisons, including privacy loss and canary exposure.

2. A new loss function that clearly divides the loss into utility and privacy.

3. This paper presents interesting findings about the privacy and utility of LoRA.

**Weaknesses:**

1. Though the analysis of privacy and utility loss is interesting, the current design of privacy loss seems to focus only on canary-related attacks, ignoring membership inference attack, which is also an important part of privacy. Therefore, the findings on the Privacy of LoRA might be overstated.

2. Some phenomenon during training should be explained. For example, in Figure 3 (b), the utility actually decreases with more training.

3. Only LoRA is considered as PEFT methods in the paper. Experiments on more PEFT methods and variants of LoRA could enhance the findings in the paper.

**Questions:**

Please see the weakness part

---

### Official Review · Reviewer_cj4R · 2025-10-31

**Soundness:** 1
**Presentation:** 3
**Contribution:** 1
**Rating:** 2
**Confidence:** 3

**Summary:**

This work investigates the privacy, utility and efficiency tradeoff of using LoRA for fine-tuning LLMs. The paper establishes new definitions for privacy and utility in terms of sensitive tokens and show that LoRA can achieve comparable privacy-utility tradeoff much more efficiently than DP-SGD. The basic privacy metric is defined as the models ability to "recollect sensitive tokens in training data" and utility is defined as the models ability to "predict nonsensitive tokens in test data". For privacy, the authors choose to capture this intuition using a privacy loss (increase in likelihood of outputting sensitive token compared to base model) metric and a more standard "canary exposure" metric. The authors evaluate 3 methods: full finetuning, LoRA fine tuning and DP-SGD full fine-tuning on four model families (Pythia, Gemma, Llama, Qwen) and two datasets (CustomerSim, SynBio) and find that FFT has poor privacy, DP-SGD is private but computationally expensive, and LoRA is private, efficient, and maintains utility. The authors also provide a theorem that compares DP-SGD and LoRA finetuning to show that they work similarly in restricting the information from each example.

**Strengths:**

- The paper addresses an important problem of understanding privacy, utility and efficiency tradeoffs of LLM finetuning.
- The paper is well written and easy to follow.
- The experiments are conducted on a wide variety of models (Pythia, Gemma, Llama, Qwen) and on 2 different datasets.

**Weaknesses:**

- DP-SGD baseline seems deeply flawed. Several issues that jump out are: no epsilon guarantee, no mention of large batch sizes, token level clipping ("...where each sample corresponds to a token...").
- Does not compare against DP-LoRA
- Privacy loss is fairly non-standard.
- Theorem 1 is trivial and does not add any insights.
- Using GPT-4 for identifying sensitive tokens seems problematic when it is a top line metric for this work.

**Questions:**

- Can the authors clarify how DP-SGD is setup and what epsilon values are considered?
- Would authors consider running an evaluation of DP-LoRA?

---

### Official Review · Reviewer_6nLf · 2025-11-01

**Soundness:** 1
**Presentation:** 2
**Contribution:** 2
**Rating:** 2
**Confidence:** 4

**Summary:**

This paper compares the empirical effects of privacy preservation (DP-SGD on the entire model) with parameter-efficient fine-tuning (LoRA, without DP). Salient aspects include new ways of empirically quantifying privacy as the loss on sensitive (as per GPT-4) tokens in the training set, and downstream utility, which is quantified as loss on non-sensitive (as per GPT-4) tokens on a held-out set.

**Strengths:**

The idea of empirically computing effective privacy (in a DP sense) protection from non-DP heuristics is a common strategy. This paper takes the same approach for large language models with parameter-efficient fine-tuning as a heuristic.

The paper is well-written, well-organized, and easy to follow (although the plots are hard to interpret, more on this later).

**Weaknesses:**

[W1] **Justification of the Metrics**: Even assuming the identification of sensitive and non-sensitive tokens is perfect (more in W2), the newly introduced privacy and utility metrics are not fully justified. For example:
* The onus is on the paper to justify these metrics fully and put them in context given past work. There are several standard and well-accepted measures of privacy: exposure (which is used in the experiments), success of membership inference attacks, etc. Why is this measure better than them? Is it computationally less expensive? If yes, does it convey the same insights or does it have other drawbacks? That has be conveyed through detailed evaluations.
* In particular, I'm not convinced that the training loss on the sensitive tokens a good measure of privacy? It is very easy to inflate that loss by making a single highly confident wrong prediction. As such, lower quantiles (or even the minimum) might be a better metric.
* As for the utility, I think the metric is better justified. However, there are several task-specific measures of downstream utility that go beyond the cross entropy loss of next-token prediction: accuracy for multi-choice tasks, ROUGE for evaluation, toxicity/bias measurements, MAUVE for open-ended generation, FactScore for factuality, etc. It would be good to consider at least a small subset of them.
* If a reader is not convinced about these metrics, all the findings in the paper are questionable.


[W2] Applying this method requires identifying sensitive tokens vs non-sensitive ones (from a privacy perspective). Datasets do not come with this annotation. The authors use GPT-4 for this and verify that "75% [i.e. 30 of the 40 participants] found the GPT-4 annotations to be accurate". This is not enough information to gauge the accuracy of this labeling system, which is a fundamental core component of the proposed approach. For example:
* Missed detection: what is the rate of missed sensitive tokens?
* False identifications: how many non-sensitive tokens are identified as sensitive?
* What are some examples of the above two? Is there some way of trading off these errors? For example, since privacy evaluations are more crucial (and there can be other ways to measure utility), I would imagine that it is more realistic to minimize the missed detection.

[W3] More fundamentally, the paper mixes up two different philosophies:
* theoretical privacy protection: no attack/adversary in the world can ever infer more than "x" amount of sensitive information about the data, e.g. DP
* empirical privacy protection: here is an attack that leaks "y" amount of sensitive information. Therefore, the real leakage >= y.

There is an implicit mix up between both. Empirical privacy measurement is only as good as the attack. And since privacy leakage is usually worst-case, the attacks are typically adversarially crafted. That is not the case here.

It is possible to compare lower bounds (empirical leakage) and upper bounds (DP bounds), but it must be done carefully, acknowledging all the drawbacks of such a procedure. Running an empirical privacy audit is one such a way: https://papers.neurips.cc/paper_files/paper/2020/file/fc4ddc15f9f4b4b06ef7844d6bb53abf-Paper.pdf

[W4] As acknowledged in the paper, DP is often used with LoRA, at least with large models. So I'm very puzzled that DP-LoRA is not even considered as a baseline.

[W5] **Issues in the experimental settings**:
* The datasets are tiny, with 10k and 5k data points. DP usually requires very large batch sizes, usually around 1k or more. Thus, to get any meaningful privacy guarantees, the size of the dataset should be at least an order of magnitude larger. Also, what batch size is used in Sec 4.2? In fact, this increase in the batch size is the main contributor to the increased cost of DP training.
* What are the resulting values of epsilon in Sec 4.2? This is common practice in DP as it gives a fair comparison across different settings
* It is common practice to tune the learning rates after applying gradient clipping. Here, the gradients are clipped to a norm of $10^{-2}$, which is very small (typical value is 1 or 0.5 or thereabouts). Unless the learning rate is increased, not much learning takes place.

Further information on all these points can be found in, for example, https://arxiv.org/pdf/2303.00654

[W6] Plots are confusing. Which not have all lines share the same x/y-axes in Fig 4/5? It is almost impossible to compare plots.

[W7] The privacy loss measurement on p7 is hand-wavy and not precise. Models trained on D and D' would have n-1 statistically similar sequences in common, so Eq. 5 is definitely not an equality- it is most likely an over-estimate.

[W8] Section 6 is sloppy and not mathematically precise.

**Questions:**

* Efficiency of LoRA: the paper says that much larger batch sizes are possible. It would be good to see concrete numbers.
* Line 408: what does "best" mean? This is a multi-objective optimization between privacy, utility, computation cost.

---

### Meta-Review · Area_Chair_us3K · 2026-01-10

**Summary:**

The support for a rejection decision centers on flaws in the experimental design and validation methods in the paper. The reviewers criticized the baselines, noting that the DP-SGD implementation was suboptimal (e.g., it used insufficient batch sizes) and that the study critically omitted the DP-LoRA baseline, which is critical for a fair comparison of privacy and efficiency. Moreover, the proposed metrics for privacy and utility were deemed non-standard and unconvincing, in particular the reliance on GPT-4 for data annotation; its low accuracy raised concerns about the validity of the ground truth used for evaluation. Furthermore the work was critiqued for lacking theoretical rigor, as it conflates empirical observations of privacy with formal guarantees required to substantiate the claims. In its present form, the paper doesn't meet the bar to appear at ICLR.

**Reviewer Concerns:**

Not applicable since no rebuttal was provided.

**Reviewer Scores:**

Not applicable since no rebuttal was provided.

---

### Decision · Program_Chairs · 2026-01-26

Reject